# Continual learning: a feature extraction formalization, an efficient algorithm, and barriers

**Binghui Peng**
Columbia University
bp2601@columbia.edu

**Andrej Risteski**
Carnegie Mellon University
aristesk@andrew.cmu.edu

## Abstract

Continual learning is an emerging paradigm in machine learning, wherein a model is exposed in an online fashion to data from multiple different distributions (i.e. environments), and is expected to adapt to the distribution change. Precisely, the goal is to perform well in the new environment, while simultaneously retaining the performance on the previous environments (i.e. avoid "catastrophic forgetting"). While this setup has enjoyed a lot of attention in the applied community, there hasn't be theoretical work that even formalizes the desired guarantees. In this paper, we propose a framework for continual learning through the framework of feature extraction—namely, one in which features, as well as a classifier, are being trained with each environment. When the features are linear, we design an efficient gradient-based algorithm DPGrad, that is guaranteed to perform well on the current environment, as well as avoid catastrophic forgetting. In the general case, when the features are non-linear, we show such an algorithm cannot exist, whether efficient or not.

## 1 Introduction

In the last few years, there has been an increasingly large focus in the modern machine learning community on settings which go *beyond iid data*. This has resulted in the proliferation of new concepts and settings such as out-of-distribution generalization [16], domain generalization [3], multi-task learning [41], continual learning [25] and etc. *Continual learning*, which is the focus of this paper, concerns learning through a sequence of environments, with the hope of retaining old knowledge while adapting to new environments.

Unfortunately, despite a lot of interest in the applied community—as evidenced by a multitude of NeurIPS and ICML workshops [26, 12, 30]—approaches with formal theoretical guarantees are few and far between. The main reason, similar encountered as its cousin fields like out-of-distribution generalization or multi-tasks learning, usually come with some "intuitive" desiderata — but no formal definitions. What's worse, it's often times clear that without strong data assumptions—the problem is woefully ill-defined.

The intuitive desiderata the continual learning community has settled on is that the setting involves cases where an algorithm is exposed (in an online fashion) to data sequentially coming from different distributions (typically called "environments", inspired from a robot/agent interacting with different environments). Moreover, the goal is to keep the size of the model being trained fixed, and make sure the model performs well on the current environment *while simultaneously* maintaining a good performance in the previously seen environments. In continual learning parlance, this is termed "resistance to catastrophic forgetting".

It is clear that some of the above desiderata are shared with well-studied learning theory settings (e.g. online learning, lifelong learning), while some aspects differ. For example, in online learning,

36th Conference on Neural Information Processing Systems (NeurIPS 2022).

we don't care about catastrophic forgetting (or we only do so in some averaged sense); in lifelong learning, it's not necessary to keep the size of the model fixed. It is also clear that absent some assumptions on the data and the model being trained, these desiderata cannot possibly be satisfied: why would there even exist a model of some fixed size that performs well on both past environments, and current ones — let alone one that gets updated in an online fashion.

**A feature-extraction formalization of continual learning**  Our paper formalizes a setting for continual learning through the lens of *feature extraction*: the model maintains a *fixed* number of (trainable) features, as well as a *linear classifier* on top of said features. The features are updated for every new environment, with the objective that the features are such that a good linear classifier exists for the *new* environment, while the previously trained linear classifiers (on the updated features) are still good for the past environments. The reason the linear classifiers from previous rounds are not allowed to be updated is storage efficiency: in order to tune the prompts, one needs to store the training data from previous tasks, this would bring a storage overhead and potentially privacy concerns. This is a very common approach in practice—examples of this are systems involving large amounts of data of a streaming nature (e.g. Google searches, Youtube, a robotic agent interacting with a continual stream of environments), and it be would prohibitive to store it for later fine tuning. The number of features is kept fixed for the same reason: if we are to expand with new features for every new environment, the model size (and hence storage requirements) would grow.

We prove two main results for our setting.

1. When the features are a linear function of the input data, and a good set of features exist, we design *an efficient algorithm*, named doubly projected gradient descent, or DPGrad, that has a good accuracy on all environments, and resists catastrophic forgetting. Our algorithm, while being novel, bears some resemblance to a class of projection-based algorithms used in practice [11, 5] – we hope the theoretical analysis can shed insight onto large scale continual learning.

2. When the features are allowed to be a non-linear function of the input, we show that continual learning is not possible—in general. Namely, we construct an instance for which even if a good set of features exists, the online nature of the setting, as well as the fact that the linear classifiers for past environments are not allowed to be updated, makes it possible for the algorithm to "commit" to linear classifiers, such that either catastrophic forgetting, or poor performance on the current environment has to occur.

## 2  Our results

### 2.1  Problem formulation

In a continual learning problem, the learner has sequential access to $k$ environments. In the $i$-th ($i \in [k]$) environment, the data is drawn i.i.d. from the underlying distribution $\mathcal{D}_i$ over $\mathbb{R}^d \times \mathbb{R}$, denoted as $(x, y) \sim \mathcal{D}_i$, where $x \in \mathbb{R}^d$ is the input and $y \in \mathbb{R}$ is the label. Motivated by the empirical success of representation learning [2, 9], we formulate the continual learning problem through the feature extraction view: The learner is required to learn a common feature mapping (also known as representation function) $R : \mathbb{R}^d \to \mathbb{R}^r$ that maps the input data $x \in \mathbb{R}^d$ to a low dimensional representation $R(x) \in \mathbb{R}^r$ ($r \ll d$), along with a sequence of task-dependent linear classifiers (also known as linear prompts) $v_1, \ldots, v_k \in \mathbb{R}^r$ on top of the representation. Precisely, the prediction of the $i$-th environment is made by $f(x) = \langle v_i, R(x) \rangle$.

As this is the first-cut study, we focus on the *realizable* and the *proper* learning setting.[1] That is, we assume the existence of a feature mapping $R$ in the function class $\mathcal{H}$ (which is known in advance) and a sequence of linear predictor $v_1, \ldots, v_k$ such that for any $i \in [k]$ and any data $(x, y) \sim \mathcal{D}_i$, $y = \langle v_i, R(x) \rangle$ (realizable). The learner is required to output a function $R$ that belongs to the hypothesis class $\mathcal{H}$ (i.e. the learner is proper).

**Remark 2.1** (Known environment identity)**.** *Our model requires the knowledge of environment identity at test time, and thus can be classified into the category of incremental task learning. We*

---

[1]We note it is possible to extend our algorithmic result to the non-realizable setting, provided the label has symmetric sub-gaussian noise.

*note there is also empirical research focusing on unknown environment identity, which would be an interesting direction for future work (See Section 7).*

The guarantee that we wish our learning algorithm to obtain is as follows:

**Definition 2.2** (Goal of Continual Learning). *Let $d, k, r \in \mathbb{N}$, $r \ll d, k$, $\epsilon \in (0, 1/2)$. Let $\mathcal{H}$ be a function class in which the feature mappings from $\mathbb{R}^d$ to $\mathbb{R}^r$ lie. The continual learning problem involves $k$ environments $\mathcal{D}_1, \ldots, \mathcal{D}_k$. We assume there exists a function $R^\star \in \mathcal{H}$ and a sequence of linear classifiers $v_1^\star, \ldots, v_k^\star \in \mathbb{R}^r$ such that for any $(x, y) \sim \mathcal{D}_i$ ($i \in [k]$), the label satisfies $y = \langle v_i^\star, R^\star(x) \rangle$.*

*The continual learner has sequential access to environments $\mathcal{D}_1, \ldots, \mathcal{D}_k$, as well as access to arbitrarily many samples per environment[2]. The goal is to learn a representation function $R \in \mathcal{H}$ and a sequence of linear prompts $v_1, \ldots, v_k \in \mathbb{R}^r$ that achieve a good accuracy on the current task and do not suffer from catastrophic forgetting. Formally, in the $i$-th environment ($i \in [k]$), the learner optimizes the feature mapping $R$ and the linear classifier $v_i$ (without changing $v_1, \ldots, v_{i-1}$) and aims to satisfy*

- **Avoid catastrophic forgetting***: During the execution of the $i$-th task, the algorithm guarantees that*

$$L(R, v_j) := \frac{1}{2} \mathbb{E}_{(x,y) \sim \mathcal{D}_j} (\langle v_j, R(x) \rangle - y)^2 \leq \epsilon \ \ \text{for all } j = 1, \ldots, i - 1,$$

- **Good accuracy on the current task:** *At the end of $i$-th task, the algorithm guarantees that*

$$L(R, v_i) := \frac{1}{2} \mathbb{E}_{(x,y) \sim \mathcal{D}_i} (\langle v_i, R(x) \rangle - y)^2 \leq \epsilon.$$

*For linear feature mapping, the representation function can be written in a linear form $R(x) = U^\top x$ for some $U \in \mathbb{R}^{d \times r}$, and it implies the $i$-th environment is generated by a linear model. That is, defining $w_i = U v_i \in \mathbb{R}^d$, one can write $y = \langle v_i, U^\top x \rangle = \langle w_i, x \rangle$.*

**Remark 2.3** (The benefit of continual learning with linear feature). *Note, for linear features, it's in principle possible to just learn a sequence of linear classifiers $w_1, \ldots, w_k \in \mathbb{R}^d$ separately— without learning a low-dimensional featurizer. Choosing an $r$-dimensional featurizer confers memory efficiency ($O(kr + dr)$ vs. $O(dk)$) and sample efficiency ($O(r)$ vs. $O(d)$ samples per task in the asymptotic regime $k \to \infty$). Furthermore, the linear case is a sandbox that can be mathematically analyzed and can generate insights for the nonlinear case as well.*

## 2.2 DPGrad: Efficient gradient based method for linear features

For the case of linear features, we propose an efficient algorithm which we term DPGrad (pseudocode in Algorithm 1), which is an efficient gradient based method and provably learns the representation while avoids catastrophic forgetting. Towards stating the result, we make a few technical assumptions.

**Assumption 2.4** (Distribution assumption). *For any $i \in [k]$, we assume $\mathcal{D}_i$ has zero means and it is in isotropic position, that is, $\mathbb{E}_{x \sim \mathcal{D}_i}[x] = \vec{0}$ and $\mathbb{E}_{x \sim \mathcal{D}_i}[xx^\top] = I$.*

**Remark 2.5.** *This assumption is largely for convenience. In fact, one can replace the isotropic condition with a general bounded covariance assumption, our algorithm still can work with extra preprocessing step, and the sample complexity scales with the condition number of covariance matrix.*

**Assumption 2.6** (Range assumption). *For any $i \in [k]$, $w_i$ has bounded norm, i.e., $\|w_i\|_2 \leq D$.*

**Assumption 2.7** (Signal assumption). *For any $i \in [k]$, let $\mathsf{W}_i = \mathsf{span}(w_1, \ldots, w_i)$, $\mathsf{W}_{i,\perp}$ be the space perpendicular to $\mathsf{W}_i$ and $P_{\mathsf{W}_i}, P_{\mathsf{W}_{i,\perp}}$ be the projection operator. We assume either $w_i$ belongs to $\mathsf{W}_{i-1}$ or it has non-negligible component orthogonal to $\mathsf{W}_{i-1}$, i.e., $\|P_{\mathsf{W}_{i-1,\perp}} w_i\|_2 \in \{0\} \cup [1/D, D]$.*

**Assumption 2.8** (Bit complexity assumption). *Each coordinate of $w_i$ is a multiple of $\nu > 0$.*

**Remark 2.9.** *The Range and Signal assumptions are standard in the statistical learning literature. The former ensures an upper bound on $\|w_i\|_2$ and the later ensures that a new task provides enough "signal" for new features. They are used to set up learning rate and number of gradient iterations.*

---

[2]The results easily extend to the finite sample case using standard techniques. We focus on the population results to keep the focus on the online nature of the environments.

**Remark 2.10.** *The Bit complexity assumption states that $w_i$ can be described with a finite number of bits, and is mostly for convenience — namely so we can argue we exactly recover $w_i$—which makes calculations involving projections of features learned in the past cleaner. Since the number of gradient iterations only depends* logarithmically *on $\nu$, one can relax the bit complexity restriction to only approximately recovering the ground truth features up to a polynomially small (in $d, k, D, \epsilon$) precision. Our argument can still go through at the cost of some additional error analysis.*

The main result is then as follows:

**Theorem 2.11** (Continual learning with linear features). *Let $k, d, r \in \mathbb{N}$, $r \ll k, d$, $\epsilon \in (0, 1/2)$. When the features are a linear function over the input data, under Assumption 2.4 and Assumption 2.6-2.8, with high probability,* DPGrad *provably achieves good loss and avoids catastrophic forgetting. In particular, during the execution of $i$-th environment,* DPGrad *always ensures*

$$L(U, v_j) := \frac{1}{2} \mathop{\mathbb{E}}_{(x,y) \sim \mathcal{D}_j} (x^\top U v_j - y)^2 \leq \epsilon, \quad \text{for all } j = 1, 2, \ldots, i - 1, \tag{1}$$

*and at the end of $i$-th environment,* DPGrad *ensures*

$$L(U, v_i) = \frac{1}{2} \mathop{\mathbb{E}}_{(x,y) \sim \mathcal{D}_i} (x^\top U v_i - y)^2 \leq \epsilon. \tag{2}$$

### 2.3 Fundamental obstructions for non-linear features

The continual learning setup with non-linear features turns out to be much more difficult — even without computational constraints. Our result rules out the existence of a proper continual learner, even when all environment distributions are uniform and the representation function is realizable by a two-layer convolutional neural network.

**Theorem 2.12** (Barrier for Continual learning with non-linear feature). *Let $k, r \geq 2, d \geq 3$. There exists a class of non-linear feature mappings and a sequence of environments, such that there is no (proper) continual learning algorithm that can guarantee to achieve less than $\frac{1}{1000}$-error over all environments with probability at least $1/2$, under the feature extraction formalization of Definition 2.2.*

## 3 Related work

**Continual learning in practice**    The study of continual learning (or lifelong learning) dates back to the work of [36] and it receives a surge of research interest over recent years [15, 19, 11, 5, 14, 31, 34, 17, 29, 39, 18]. A central challenge in the field is to avoid *catastrophic forgetting* [24, 23], which the work of [15] observed happened for gradient-based training of neural networks. While there is a large amount of empirical work, we'll briefly summarize the dominant approaches. The *regularization based approach* alleviates catastrophic forgetting by posing constraints on the update of the neural weights. The elastic weight consolidation (EWC) approach [19] adds weighted $\ell_2$ regularization to the objective function that penalizes the movement of neural weights. The *orthogonal gradient descent* (OGD) algorithm from [11, 5] enforces the gradient update being orthogonal to update direction (by viewing the gradients as a high dimensional vector). The *memory replay approach* restores data from previous tasks and alleviates catastrophic forgetting by rehearsing in the later tasks. [31] introduces experience replay to continual learning. [14] trains a deep generative model (a.k.a. GAN) to simulate past dataset for future use. The *dynamic architecture approach* dynamically adjusts the neural network architecture to incorporate new knowledge and avoid forgetting. The progressive neural network [34] blocks changes to the existing network and expands the architecture by allocating a new subnet to be trained with the new information. We refer the interested reader to more complete surveys [25, 7].

**Continual learning in theory**    In comparison to the vast empirical literature, theoretical works are comparatively few. The work of [6] characterize the memory requirement of continual learning, when the environment identity is unknown. The works [35, 27, 1, 4] provide sample complexity guarantees on lifelong learning. Their approaches can be categorized roughly into the *duplicate and fine-tuning* paradigm: The algorithm maintains a weighted combination over a family of representation functions and the focus is on the sample complexity guarantee. By contrast, we focus on the *feature extraction* paradigm and learn linear prompts on top of a *single* representation function. Both the duplicate-and-fine-tuning and the feature extraction paradigm have been extensively investigated in the literature,

detailed discussions can be found at [7] and we provide a brief comparison. From an algorithmic perspective, learning a weighted combination over a family of representation functions (i.e. the duplicate and fine-tuning) is much easier, as one can always initiates a new representation function for a new task. The algorithmic convenience allows previous literature focus more on the generalization and sample complexity guarantee, culminating with the recent work of [4]. We note again that learning a single representation function and task specific linear prompts is much more challenging, but has practical benefits, e.g. memory efficiency. For example, in the applications of NLP, the basic representation function (e.g. BERT [9]) is already overparameterized and contains billions of parameters. It is then formidable to maintain a large amount of the basic models and learn a linear combination over them. We mention several more works that are morally related in Appendix A.

## 4 Continual learning with linear feature

We restate our main result for linear feature mapping.

**Theorem 2.11** (Continual learning with linear features). *Let $k, d, r \in \mathbb{N}$, $r \ll k, d$, $\epsilon \in (0, 1/2)$. When the features are a linear function over the input data, under Assumption 2.4 and Assumption 2.6-2.8, with high probability,* DPGrad *provably achieves good loss and avoids catastrophic forgetting. In particular, during the execution of $i$-th environment,* DPGrad *always ensures*

$$L(U, v_j) := \frac{1}{2} \mathop{\mathbb{E}}_{(x,y) \sim \mathcal{D}_j} (x^\top U v_j - y)^2 \le \epsilon, \quad \text{for all } j = 1, 2, \ldots, i-1, \tag{1}$$

*and at the end of $i$-th environment,* DPGrad *ensures*

$$L(U, v_i) = \frac{1}{2} \mathop{\mathbb{E}}_{(x,y) \sim \mathcal{D}_i} (x^\top U v_i - y)^2 \le \epsilon. \tag{2}$$

### 4.1 Algorithm

A complete and formal description of DPGrad is presented in Algorithm 1. DPGrad simultaneously updates the matrix of features $U$, as well as the linear classifier $v_i$ using gradient descent—with the restriction that the update of $U$ only occurs along directions that are orthogonal to the column and row span of the previous feature matrix. Intuitively, one wishes the projection guarantees that existing features that have been learned are not erased or interfered by new environments. Due to the quadratic nature of the loss, and the appearance of "cross-terms", this turns out to require both column and row orthogonality, and interestingly deviates from the practically common OGD method [11, 5].

In more detail, at the beginning of the $i$-th ($i \in [k]$) environment, DPGrad adds Gaussian noise to the feature matrix $U$ and the linear classifier $v_i$, to generate a good initialization for $U$ and $v_i$. Subsequently, we perform gradient descent to both the feature mapping matrix $U$ and linear classifier $v_i$—except $U$ is only updated along orthogonal directions w.r.t. the column span and the row span. At the end of each environment, DPGrad has a post-processing step to recover the ground truth $w_i$ by rounding each entry of $U v_i$ to the nearest multiple of $\nu$,[3] and then update the column and row span if the orthogonal component is non-negligible. The reason for the later step is that we only need to preserve row space when encountering new features.

**Parameters** We use $\sigma$ to denote the initialization scale, $\eta$ to denote the learning rate, and $T$ to denote the number of iterations for each task. These are all polynomially small parameters, whose scaling is roughly $D, d, k \ll \sigma^{-1} \ll \eta^{-1} < T$.

**Notation** We write $[n] = \{1, 2, \ldots, n\}$, $[n_1 : n_2] = \{n_1, \ldots, n_2\}$. We use $\mathsf{rand}(n_1, n_2) \in \mathbb{R}^{n_1 \times n_2}$ to denote a size $n_1 \times n_2$ matrix whose entries are draw from random Gaussian $\mathsf{N}(0, 1)$. For each $i \in [k]$, $t \in [0 : T]$, denote $U_{i,t}$ to be the feature matrix in the $t$-th iteration of the $i$-th environment (after performing the gradient update), denote $v_{i,t}$ similarly. DPGrad includes a projection step at the end of $i$-th environment, we use $U_{i,\mathsf{end}}$ to denote the feature matrix after this projection. We use $\mathsf{W}_i$ (resp. $\mathsf{V}_i$) to denote the column (resp. row) space maintained at the end of $i$-th environment. Let $\mathsf{W}_\perp \subseteq \mathbb{R}^n$ be the subspace orthogonal to $\mathsf{W}$ and define $\mathsf{V}_\perp$ similarly. Let $P_\mathsf{W}, P_\mathsf{V}, P_{\mathsf{W}_\perp}, P_{\mathsf{V}_\perp}$ be the projection onto $\mathsf{W}, \mathsf{V}, \mathsf{W}_\perp, \mathsf{V}_\perp$ separately.

---

[3]This is the only place where we use the Bit complexity assumption.

**Algorithm 1** Doubly projected gradient descent (DPGrad)

---
1: $\mathsf{W} \leftarrow \emptyset, \mathsf{V} \leftarrow \emptyset, U \leftarrow \mathbf{0}$ ▷ $U \in \mathbb{R}^{d \times r}$
2: $\sigma \leftarrow \widetilde{O}(\frac{\epsilon}{d^2 k D^4}), \eta \leftarrow O(\frac{\sigma^3}{k^2 D^5}), T \leftarrow O(\frac{D}{\eta} \log \frac{Dkd}{\epsilon \nu}) + O(\frac{D}{\eta} \log \frac{k}{\sigma})$
3: **for** $i = 1, \ldots, k$ **do**
4: $\quad U_{\text{init}} \leftarrow \sigma \cdot P_{\mathsf{W}_\perp} \mathsf{rand}(d, r) P_{\mathsf{V}_\perp}, v_i \leftarrow \sigma \cdot \mathsf{rand}(r)$ ▷ $U_{\text{init}} \in \mathbb{R}^{d \times r}, v_i \in \mathbb{R}^r$
5: $\quad U \leftarrow U + U_{\text{init}}$
6: $\quad$ **for** $t = 1, \ldots, T$ **do**
7: $\qquad \nabla_U \leftarrow \mathbb{E}_{(x,y) \sim \mathcal{D}_i}[x(x^\top U v_i - y) v_i^\top], \nabla_{v_i} \leftarrow \mathbb{E}_{(x,y) \sim \mathcal{D}_i}[U^\top x(x^\top U v_i - y)]$
8: $\qquad U = U - \eta P_{\mathsf{W}_\perp} \nabla_U P_{\mathsf{V}_\perp}$
9: $\qquad v_i = v_i - \eta \nabla_{v_i}$
10: $\quad$ **end for**
11: **end for**
12: $\widehat{w}_i \leftarrow \mathsf{Round}_\nu(U v_i)$ ▷ Round to the nearest multiple of $\nu$, $\widehat{w}_i \in \mathbb{R}^d$
13: **if** $\|P_{\mathsf{W}_\perp} \widehat{w}_i\|_2 \geq 1/D$ **then** $\mathsf{W} \leftarrow \mathsf{span}(\mathsf{W} \cup \widehat{w}_i), \mathsf{V} \leftarrow \mathsf{span}(\mathsf{V} \cup v_i)$
14: $U \leftarrow P_{\mathsf{W}} U P_{\mathsf{V}}$

---

### 4.2 Analysis

We sketch the analysis of DPGrad and prove Theorem 2.11. Due to space limitation, the detailed proof is deferred to Appendix B. The proof proceeds in the following four steps:

1. The first step, presented in Section 4.2.1, reduces continual learning to a problem of continual matrix factorization and it allows us to focus on a more algebraically friendly objective function.

2. We then present some basic linear-algebraic facts to decompose the feature mapping matrix $U$, its gradient, and the loss into orthogonal components. The orthogonality of gradient update allows us to decouple the process of *leveraging the existing features* and the process of *learning a new feature*, as reflected in the loss terms and gradient update rules. See Section 4.2.2 for details.

3. In Section 4.2.3, we zoom into one single environment, and prove DPGrad provably converges to a global optimum, assuming the feature matrix $U$ from previous environment is well conditioned. This step contains the major bulk of our analysis: The objective function of continual matrix factorization is non-convex, and no regularization or spectral initialization used. (We cannot re-initialize, lest we destroy progress from prior environments.)

4. Finally, in Section 4.2.4, we inductively prove that DPGrad converges and the feature matrix is always well-conditioned. This wraps up the entire proof.

#### 4.2.1 Reduction

We first recall the formal statement of the problem of continual matrix factorization.

**Definition 4.1** (Continual matrix factorization). *Let $d, k, r \in \mathbb{N}$, $r \ll d, k$, $\epsilon > 0$. Let $W = [w_1, \ldots, w_k] = U^\star (V^\star)^\top \in \mathbb{R}^{d \times k}$, where $U^\star \in \mathbb{R}^{d \times r}, V^\star \in \mathbb{R}^{k \times r}$. In an continual matrix factorization problem, the algorithm receives $w_i \in \mathbb{R}^d$ in the $i$-th step, and it is required to maintain a matrix $U \in \mathbb{R}^{d \times r}$ and output a vector $v_i \in \mathbb{R}^r$ such that*

$$\widehat{L}(U, v_i) = \frac{1}{2}\|U v_i - w_i\|_2^2 \leq \epsilon, \tag{3}$$

*and*

$$\widehat{L}(U, v_j) = \frac{1}{2}\|U v_j - w_j\|_2^2 \leq \epsilon \quad j = 1, \ldots, i-1. \tag{4}$$

The key observation is that running DPGrad on the original continual learning objective (2) implicitly optimizes the continual matrix factorization objective (3) (Lemma 4.2). Moreover, an $\epsilon$-approximate solution of continual matrix factorization is also an $\epsilon$-approximate solution of continual learning (Lemma 4.3).

**Lemma 4.2** (Gradient equivalence). *Under Assumption 2.4, for any $i \in [k]$, the gradient update of objective* (2) *equals the gradient update of objective* (3).

**Lemma 4.3** (Objective equivalence). *For any $w_1, \ldots, w_k \in \mathbb{R}^d$, $U \in \mathbb{R}^{d \times r}$ and $v_1, \ldots, v_k \in \mathbb{R}^r$, suppose $\widehat{L}(U, v_i) = \frac{1}{2}\|Uv_i - w_i\|_2^2 \leq \epsilon$ holds for all $i \in [k]$, then $L(U, v_i) = \frac{1}{2}\mathbb{E}_{(x,y) \sim \mathcal{D}_i}(x^\top U v_i - y)^2 \leq \epsilon$.*

Combining the above observations, it suffices to analyse DPGrad for continual matrix factorization and prove Eq. (3) and Eq. (4).

### 4.2.2 Decomposition

We first provide some basic linear algebraic facts about orthogonal decompositions. For any $i \in [k]$, we decompose $U_i, v_i, w_i$ along $\mathsf{W}_{i-1}, \mathsf{W}_{i-1,\perp}, \mathsf{V}_{i-1}$ and $\mathsf{V}_{i-1,\perp}$.

Let $w_i = w_{i,A} + w_{i,B}$ where $w_{i,A} \in \mathsf{W}_{i-1}$ and $w_{i,B} \in \mathsf{W}_{i-1,\perp}$. Note this decomposition is unique. We focus on the case that $\|w_{i,B}\|_2 \in [1/D, D]$ in the following statements, and the case of $\|w_{i,B}\|_2 = 0$ carries over easily. (These are the only two cases, per Assumption 2.7). Similarly, let $U_i = U_{i,A} + U_{i,B}$, where each column of $U_{i,A}$ lies $\mathsf{W}_{i-1}$ and each column of $w_{i,B}$ lies in $\mathsf{W}_{i-1,\perp}$. (Note, again, $U_{i,A}$ and $U_{i,B}$ are unique.) We further write $U_{i,B} = w_{i,B}x_i^\top + U_{i,2}$, where the columns of $U_{i,2}$ lie in $\mathsf{W}_{i-1,\perp} \backslash \{w_{i,B}\}$. Finally, denote $v_i = v_{i,1} + v_{i,2}$ with $v_{i,1} \in \mathsf{V}_{i-1}$ and $v_{i,2} \in \mathsf{V}_{i-1,\perp}$. We summarize decompositions mentioned above, with a few additional observations, in the lemma below:

**Lemma 4.4** (Orthogonal decomposition). *For any $i \in [k]$ and any $t \in [0 : T]$, there exists an unique decomposition of $U_{i,t}, w_i$ and $v_{i,t}$ of the form*

$$U_{i,t} = U_{i,A,0} + U_{i,B,t}, \qquad \mathsf{column}(U_{i,A,0}) \in \mathsf{W}_{i-1}, \mathsf{column}(U_{i,B,t}) \in \mathsf{W}_{i-1,\perp},$$
$$\mathsf{row}(U_{i,A,0}) \in \mathsf{V}_{i-1}, \mathsf{row}(U_{i,B,t}) \in \mathsf{V}_{i-1,\perp}$$
$$w_i = w_{i,A} + w_{i,B}, \qquad w_{i,A} \in \mathsf{W}_{i-1}, w_{i,B} \in \mathsf{W}_{i-1,\perp}$$
$$U_{i,B,t} = w_{i,B}x_{i,t}^\top + U_{i,2,t}, \qquad x_{i,t} \in \mathsf{V}_{i-1,\perp}, \mathsf{row}(U_{i,2,t}) \in \mathsf{V}_{i-1,\perp}, w_{i,B} \perp \mathsf{column}(U_{i,2,t})$$
$$v_{i,t} = v_{i,1,t} + v_{i,2,t} \qquad v_{i,1,t} \in \mathsf{V}_{i-1}, v_{i,2,t} \in \mathsf{V}_{i-1,\perp}.$$

*Here we use* $\mathsf{column}(A), \mathsf{row}(A)$ *to denote the column and row space of matrix $A$, and $\mathsf{column}(A) \in \mathsf{W}$ if the column space of $A$ is a subspace of $\mathsf{W}$.*

Since $U_{i,A,t}$ remains unchanged for $t = [0 : T]$, we abbreviate it as $U_{i,A}$ hereafter. We next provide the exact gradient update of each component under loss function $\widehat{L}(U_i, v_i) = \frac{1}{2}\|U_i v_i - w_i\|_2^2$ and orthogonal projection.

**Lemma 4.5** (Gradient formula). *For any $i \in [k]$, the gradient update (after projection) obeys the relations (1) $\nabla_{x_i}(\widehat{L}) = v_{i,2}(x_i^\top v_{i,2} - 1)$; (2) $\nabla_{U_{2,i}}(\widehat{L}) = U_{i,2}v_{i,2}v_{i,2}^\top$; (3) $\nabla_{v_{i,1}}(\widehat{L}) = U_{i,A}^\top U_{i,A}v_{i,1} - U_{i,A}^\top w_{i,A}$ and (4) $\nabla_{v_{i,2}}(\widehat{L}) = \|w_{i,B}\|_2^2(x_i^\top v_{i,2} - 1)x_i + U_{i,2}^\top U_{i,2}v_{i,2}$.*

We perform a similar decomposition to the loss function.

**Lemma 4.6** (Loss formula). *For any $i \in [k], t \in [T]$, we have*

$$\widehat{L}(U_{i,t}, v_{i,t}) = \frac{1}{2}\|U_{i,A}v_{i,1,t} - w_{i,A}\|_2^2 + \frac{1}{2}\|w_{i,B}\|_2^2(x_{i,t}^\top v_{i,2,t} - 1)^2 + \frac{1}{2}\|U_{i,2,t}v_{i,2,t}\|_2^2. \quad (5)$$

**Decoupling existing features from "new" features** We now offer some intuitive explanation for the decomposition. The first loss term in Eq. (5) quantifies the error with already learned features. That is, the matrix $U_{i,A}$ stores existing features that have been learned, and it remains unchanged during the execution of the $i$-th environment; it remains to optimize $v_{i,1,t}$ such that $U_{i,A}v_{i,1,t}$ matches $w_{i,A}$. The second and last loss term quantify the loss on a new feature, where $w_{i,B}$ is the new feature component, and the matrix $U_{i,2,t}$ can be thought of as random noise. Intuitively, one should hope $x_{i,t}^\top v_{i,2,t} = 1$ and this matches the new component of $w_{i,B}$. At the same time, one hopes $U_{i,2,t}$ would disappear, or at least, $\|U_{i,2,t}v_{i,2,t}\|_2 \to 0$ when $t \to \infty$.

### 4.2.3 Convergence

For a fixed environment, we prove w.h.p. DPGrad converges and the loss approaches to zero, given the initial feature mapping matrix $U_{i,A}$ is well conditioned.

**Lemma 4.7.** *For any $i \in [k]$, suppose $U_{i,A}$ satisfies $\frac{1}{2\sqrt{D}} \leq \sigma_{\min}(U_{i,A}) \leq \sigma_{\max}(U_{i,A}) \leq 2\sqrt{D}$, where $\sigma_{\min}(U_{i,A})$ and $\sigma_{\max}(U_{i,A})$ denote the minimum and maximum non-zero singular value of matrix $U_{i,A}$. After $T = O(\frac{D}{\eta} \log \frac{Dkd}{\epsilon\nu}) + O(\frac{D}{\eta} \log \frac{k}{\sigma})$ iterations, with probability at least $1 - O(1/k)$, the loss $\widehat{L}(U_i, v_i) \leq \epsilon\nu/Dnk$.*

**Outline of the proof**    DPGrad ensures existing features are preserved and it only optimizes the linear classifier, hence a linear convergence rate can be easily derived for the first loss term, given the feature matrix is well-conditioned. The key part is controlling the terms that capture learning with new features, i.e., the second and last loss term, where both the feature mapping $U_{i,B}$ and linear prompt $v_i$ get updated. In this case, the objective is non-convex and non-smooth. Our analysis draws inspiration from the recent work of [40], and divides the optimization process into two stages. We prove DPGrad first approaches to a nice initialization position with high probability, and then show linear convergence.[4]

To be concrete, in the first stage, we prove (1) $x_{i,t}^\top v_{i,2,t}$ moves closer to 1, and (2) $\|x_{i,t} - \|w_{i,B}\|_2 v_{i,2,t}\|_2 \approx 0$. That is, the second loss term of Eq. (5) decreases to a small constant while the pairs $x_{i,t}, v_{i,2,t}$ remain balanced and roughly equal up to scaling. Meanwhile, we note $U_{i,2,t}$ is non-increasing, though the last loss term could still increase because $\|v_{i,2,t}\|_2$ increases. In the second stage, we prove by induction that $\|U_{i,2,t}^\top v_{i,2,t}\|_2$ and $|x_{i,t}^\top v_{i,2,t} - 1|$ decay with a linear rate (hence converging to a global optimal), and $\|x_{i,t} - \|w_{i,B}\|_2 v_{i,2,t}\|_2 \approx 0$.

### 4.2.4 Induction

Lemma 4.7 proves rapid convergence of DPGrad for one single environment. To extend the argument to the whole sequence of environments, we need to ensure (1) the feature matrix is always well-conditioned and (2) catastrophic forgetting does not happen. For (1), we need to analyse the limiting point of DPGrad (there are infinitely many optimal solutions to Eq. (3)), make sure it is well-balance and orthogonal to previous row/column space. For (2), we make use of the orthogonality of DPGrad.

*Proof Sketch of Theorem 2.11.*   Thanks to the reduction established in Section 4.2.1, it suffices to prove Eq. (3) and Eq. (4). For each environment $i$ ($i \in [k]$), we inductively prove (1) DPGrad achieves good accuracy on the current environment, i.e., $\|U_{i,T} v_i - w_i\|_2 \leq \epsilon\nu$; (2) The feature matrix $U_i$ remains well conditioned, i.e. $\frac{1}{2\sqrt{D}} \leq \sigma_{\min}(U_{i,\text{end}}) \leq \sigma_{\max}(U_{i,\text{end}}) \leq 2\sqrt{D}$ and (3) The algorithm does not suffer from catastrophic forgetting, i.e., $\|U_{i,t} v_j - w_i\|_2 \leq \epsilon$ for any $j < i, t \in [T]$.

The first claim is already implied by Lemma 4.7. For the second claim, one first shows DPGrad exactly recovers $w_i$ by taking $w_i = \widehat{w}_i = \text{Round}_\nu(U_{i,T} v_i)$. When $w_{i,B} = 0$, one can prove the feature matrix does not change, i.e, $U_{i,\text{end}} = U_{i-1,\text{end}}$; when $w_{i,B} \in [1/D, D]$, then one can show $U_{i,\text{end}} \approx U_{i,\text{end}} + \frac{1}{\|v_{i,2,T}\|_2^2} w_{i,B} v_{i,2,T}^\top$, as $w_{i,B} \perp \text{column}(U_{i-1,\text{end}}), v_{i,2,T} \perp \text{row}(U_{i-1,\text{end}})$ and $\|\frac{1}{\|v_{i,2,T}\|_2^2} w_B v_{i,2,T}^\top\| \leq O(\sqrt{D})$, the feature matrix $U$ remains well-conditioned. The last claim can be derived from the orthogonality. This wraps up the proof of Theorem 2.11.    □

## 5   Lower bound for non-linear features

We next consider continual learning under a non-linear feature mapping. Learning with non-linear features turns out to be much more difficult, and our main result is to rule out the possibility of a (proper) continual learner. We restate the formal statement. The detailed proof are deferred to Appendix C.

---

[4]We note most existing works on matrix factorization or matrix sensing either require some fine-grained initialization (e.g. spectral initialization [8]) or adding a regularization term that enforces smoothness [13], none of which are applicable in our setting.

**Theorem 2.12** (Barrier for Continual learning with non-linear feature). *Let $k, r \geq 2, d \geq 3$. There exists a class of non-linear feature mappings and a sequence of environments, such that there is no (proper) continual learning algorithm that can guarantee to achieve less than $\frac{1}{1000}$-error over all environments with probability at least $1/2$,under the feature extraction formalization of Definition 2.2.*

Our lower bound is constructed on a simple family of two-layer convolutional neural network with quadratic activation functions. The input distribution is assumed to be uniform and the target function is a polynomial over the input. The first environment is constructed such that multiple global optimum exist (hence the optimization task is under-constrained). However, if a wrong optimum solution is picked, when the second environment is revealed, the non-linearity makes it impossible to switch back-and-forth.

*Proof Sketch.* It suffices to take $k = 2, d = 3, r = 2$ and we sketch the construction here. For both environments, we assume the input data are drawn uniformly at random from $\mathcal{B}_3(0, 1)$, where $\mathcal{B}_3(0, 1)$ denotes the unit ball in $\mathbb{R}^3$ centered at origin. The hypothesis class $\mathcal{H}$ consists of all two-layer convolutional neural network with a single kernel of size 2 and the quadratic activation function. That is, the representation function is parameterized by $w \in \mathbb{R}^2$ and takes the form of $R_w(x) = (\langle w, x_{1:2} \rangle^2, \langle w, x_{2:3} \rangle^2) \in \mathbb{R}^2$, where $x \in \mathbb{R}^3$, $x_{i:j} \in \mathbb{R}^{j-i+1}$ is a vector consists of the $i$-th entry to the $j$-th entry of $x$.

The hard sequence of environments are drawn from the following distribution: (1) The objective function $f_1$ of the first environment is $f_1(x) = x_2^2$; (2) The objective function $f_2$ of the second environment equals $f_2(x) = x_3^2$ with probability $1/2$, and equals $f_2(x) = x_1^2$ with probability $1/2$. Note the continual learning task is realizable and one can prove no (proper) continual learning algorithm can guarantee to achieve less than $1/1000$-error on both environments with probability at least $1/2$. $\square$

Though our lower bound instance uses a polynomial activation function, this assumption is not essential – in Appendix C, we prove similar lower bounds with a ReLU activation function.

## 6 Experiments

For linear feature functions, we perform simulations on a synthetic dataset to verify the practicality of DPGrad and compare its performance with vanilla SGD and Orthogonal gradient descent (OGD), a close practical cousin of our algorithm. In our simulations, we set $d = 100$, $r = 20$, $k = 500$ and the ground truth $U^\star, V^\star$ is drawn from Gaussian. The input data are sampled from $N(0, I_d)$ and we draw 1000 samples for each task. Additional details about the setup can be found in Appendix D.

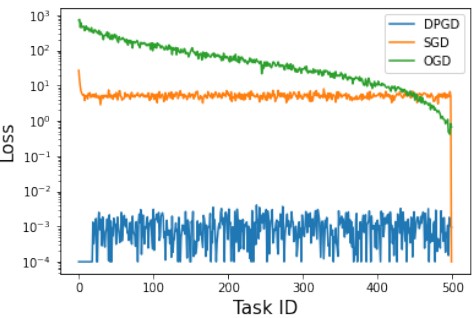

Figure 1: Continual learning with linear feature: comparative performance of DPGrad/OGD/SGD. Data is synthetically generated with $d = 100$, $r = 20$, $k = 500$ and the ground truth $U^\star, V^\star$ is drawn from Gaussian. Additional details about the setup can be found in Appendix D.

The results are presented at Figure 1. It indicates the (1) practicality of DPGrad and (2) DPGrad significantly outperforms the vanilla SGD and OGD (of course, DPGrad is designed for this kind of data). The population loss is measured at the end and the it equals $\|Uv_i - w_i\|_2$ for each task $i$. The

average error of DPGrad is $0.001$, the average error of OGD is $83.59$, the average error of SGD is $5.16$.

Moreover, in Appendix E, we provide additional experimental results on two popular benchmarks, Rotated MNIST and Permuted MNIST. Since DPGrad is designed specifically for linear regression, we provide two variants of DPGrad (without provable guarantees on their performance, of course)— one is a modification suitable for multi-class classification, the other is a modification suitable for non-linear featurizers. Detailed numbers and figures can be found in Appendix E. In brief, both algorithms alleviate catastrophic forgetting and perform much better than vanilla SGD. Furthermore, the performance of both is much more stable than OGD and the accuracy remains at a high level across tasks.

## 7    Conclusion

In this paper, we initiate a study of continual learning through *the feature extraction lens*, proposing an efficient gradient based algorithm, DPGrad, for the linear case, and a fundamental impossibility result in the general case. Our work leaves several interesting future directions. First, it would be interesting to generalize DPGrad to non-linear feature mappings (perhaps even without provable guarantees) and conduct an empirical study of its performance. Second, our impossibility result does not rule out an improper continual learner, and in general, one can always maintain a task specific representation function and achieve good performance over all environments. It would be thus interesting to investigate what are the fundamental memory-accuracy trade-offs.

## Acknowledgement

BP acknowledges support of the NSF via CCF-1909756, CCF2007443, CCF-2134105, CCF-1703925, IIS-1838154, CCF-2106429 and CCF-2107187. AR acknowledges support of the NSF via IIS-2211907, the CMU/PwC DT&I Center, and an Amazon Research Award.

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
