4:     $U_{\text{init}} \leftarrow \sigma \cdot P_{W_\perp} \text{rand}(d, r) P_{V_\perp}, v_i \leftarrow \sigma \cdot \text{rand}(r)$     $\triangleright U_{\text{init}} \in \mathbb{R}^{d \times r}, v_i \in \mathbb{R}^r$
5:     $U \leftarrow U + U_{\text{init}}$
6:     **for** $t = 1, \ldots, T$ **do**
7:        $\nabla_U \leftarrow \mathbb{E}_{(x,y) \sim \mathcal{D}_i}[x(x^\top U v_i - y) v_i^\top], \nabla_{v_i} \leftarrow \mathbb{E}_{(x,y) \sim \mathcal{D}_i}[U^\top x(x^\top U v_i - y)]$
8:        $U = U - \eta P_{W_\perp} \nabla_U P_{V_\perp}$
9:        $v_i = v_i - \eta \nabla_{v_i}$
10:     **end for**
11: **end for**
12: $\widehat{w}_i \leftarrow \text{Round}_\nu(U v_i)$        $\triangleright$ Round to the nearest multiple of $\nu$, $\widehat{w}_i \in \mathbb{R}^d$
13: **if** $\|P_{W_\perp} \widehat{w}_i\|_2 \geq 1/D$ **then** $W \leftarrow \text{span}(W \cup \widehat{w}_i), V \leftarrow \text{span}(V \cup v_i)$

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

# A    Additional related work

**Representation learning**   More broadly, our work is also closely related to representation learning. Some recent theoretical works [20, 21, 28, 38, 22, 37, 10] provide generalization and sample complexity guarantees for certain formalizations of multi-task learning based on the existence of a good representation. The work of [33, 32] formulate the problem of out-of-distribution generalization and provide theoretical guarantee, similarly, under the assumption of a good representation.

# B    Missing proof from Section 4

## B.1    Missing proof from Section 4.2.1

We first present the proof of Lemma 4.2

*Proof of Lemma 4.2.*   For any $i \in [k]$, the gradient of feature matrix $U$ w.r.t. objective Eq. (2) equals

$$\nabla_U = \mathop{\mathbb{E}}_{(x,y)\sim\mathcal{D}_i}[x(x^\top U v_i - y)v_i^\top] = \mathop{\mathbb{E}}_{x\sim\mathcal{D}_i}[x(x^\top U v_i - x^\top w_i)v_i^\top] = (Uv_i - w_i)v_i^\top. \qquad (6)$$

The first step follows from $y = x^\top w_i$ for any $(x, y) \sim \mathcal{D}_i$ and the second step follows from $\mathbb{E}_{x_i\sim\mathcal{D}_i}[xx^\top] = I_n$. The RHS of the above equation exactly equals the gradient of Eq. (3) for $U$ (before and after projection to $\mathsf{W}_{i-1}$).

We next observe

$$\nabla_{v_i} = \mathop{\mathbb{E}}_{(x,y)\sim\mathcal{D}_i}[U^\top x(x^\top U v_i - y)] = \mathop{\mathbb{E}}_{x\sim\mathcal{D}_i}[U^\top x(x^\top U v - x^\top w_i)] = U^\top(Uv_i - w_i), \qquad (7)$$

and the RHS of the above equation matches the gradient of Eq. (3) for $v_i$. We conclude the proof here.  □

We then include the proof of Lemma 4.2

*Proof of Lemma 4.3.*   We have

$$\begin{aligned}
\frac{1}{2}\mathop{\mathbb{E}}_{(x,y)\sim\mathcal{D}_i}(x^\top U v_i - y)^2 &= \frac{1}{2}\mathop{\mathbb{E}}_{(x,y)\sim\mathcal{D}_i}(x^\top U v_i - x^\top w_i)^2 \\
&= \frac{1}{2}(Uv_i - w)^\top \mathop{\mathbb{E}}_{x\sim\mathcal{D}_i}[xx^\top](Uv_i - w)^\top \\
&= \frac{1}{2}\|Uv_i - w_i\|_2^2 \le \epsilon.
\end{aligned}$$

where the first step follows from $y = x^\top w_i$ for any $(x, y) \sim \mathcal{D}_i$ and the third step follows from $\mathbb{E}_{x_i\sim\mathcal{D}_i}[xx^\top] = I_n$. This concludes the proof.  □

## B.2    Missing proof from Section 4.2.2

We first present the proof of Lemma 4.4

*Proof of Lemma 4.4.*   For the first term, when $t = 0$, one has $\mathsf{column}(U_{i,A,0}) \in \mathsf{W}_{i-1}$ and $\mathsf{column}(U_{i,B,0}) \in \mathsf{W}_{i-1,\perp}$, and these indicate (1) $U_{i,A,0} = U_{i-1,\mathsf{end}}$, $\mathsf{row}(U_{i-1,\mathsf{end}}) \in \mathsf{V}_{i-1}$ and (2) $U_{i,B,0} = U_{i,\mathsf{init}}$, $\mathsf{row}(U_{i,\mathsf{init}}) \in \mathsf{V}_{i-1,\perp}$. Hence we conclude $\mathsf{row}(U_{i,A,0}) \in \mathsf{V}_{i-1}$ and $\mathsf{row}(U_{i,A,0}) \in \mathsf{V}_{i-1,\perp}$. Since the gradient update is perform along $\mathsf{W}_{i-1,\perp}$ and $\mathsf{V}_{i-1,\perp}$, one has $U_{i,A}$ remains unchanged, i.e., $U_{i,A,t} = U_{i,A,0}$ ($t \in [T]$), and the update of $U_{i,B,t}$ is along $\mathsf{V}_{i-1,\perp}$, hence $\mathsf{row}(U_{i,B,t}) \in \mathsf{V}_{i-1,\perp}$ continues to hold.

For the third term, for any $t \in [0 : T]$, one has

$$\mathsf{V}_{i-1,\perp} \ni w_{i,B}^\top U_{i,B,t} = w_{i,B}^\top w_{i,B} x_{i,t}^\top + w_{i,B}^\top U_{i,2,t} = \|w_{i,B}\|_2^2 x_{i,t}^\top,$$

where the second step follows from $\mathsf{column}(U_{i,2,t}) \in \mathsf{W}_{i-1,\perp}\backslash\{w_{i,B}\}$. Hence we conclude $x_{i,t} \in \mathsf{V}_{i-1,\perp}$. Since $\mathsf{row}(U_{i,B,t}), \mathsf{row}(w_{i,B}x_{i,t}^\top) \in \mathsf{V}_{i-1,\perp}$, one has $\mathsf{row}(U_{i,2,t}) \in \mathsf{V}_{i-1,\perp}$.  □

We then prove

*Proof of Lemma 4.5.* The gradient of $U$ (before projection) satisfies

$$
\begin{aligned}
\nabla_{U_i} &= (U_i v_i - w_i) v_i^\top \\
&= U_{i,A} v_i v_i^\top + U_{i,B} v_i v_i^\top - w_{i,A} v_i^\top - w_{i,B} v_i^\top \\
&= (U_{i,A} v_i v_i^\top - w_{i,A} v_i^\top) + (w_{i,B} x_i^\top + U_{i,2}) v_i v_i^\top - w_{i,B} v_i^\top \\
&= (U_{i,A} v_i v_i^\top - w_{i,A} v_i^\top) + w_{i,B} v_i^\top (x_i^\top v_i - 1) + U_{i,2} v_i v_i^\top,
\end{aligned}
$$

where the first step follows from Eq. (6), the second and third steps follow from the first three terms of Lemma 4.4.

The actual update (after projection) obeys

$$
\begin{aligned}
P_{\mathsf{W}_{i-1,\perp}} \nabla_{U_i} P_{\mathsf{V}_{i-1,\perp}} &= P_{\mathsf{W}_{i-1,\perp}} ((U_{i,A} v_i v_i^\top - w_{i,A} v_i^\top) + w_{i,B} v_i^\top (x_i^\top v_i - 1) + U_{i,2} v_i v_i^\top) P_{\mathsf{V}_{i-1\perp}} \\
&= (w_{i,B} v_i^\top (x_i^\top v_i - 1) + U_2 v_i v_i^\top) P_{\mathsf{V}_{i-1\perp}} \\
&= w_{i,B} v_{i,2}^\top (x_i^\top v_{i,2} - 1) + U_{i,2} v_{i,2} v_{i,2}^\top,
\end{aligned}
$$

where the second step follows from $w_{i,A}, \mathsf{column}(U_{i,A}) \in \mathsf{W}_{i-1}$, the third step follows from $\mathsf{row}(U_{i,2}) \in \mathsf{V}_{i-1,\perp}$ and $x_i \in \mathsf{V}_{i-1,\perp}$, see Lemma 4.4 for details.

Hence, we conclude

$$
\nabla_{x_i} = v_{i,2} (x_i^\top v_{i,2} - 1) \quad \text{and} \quad \nabla_{U_{i,2}} = U_{i,2} v_{i,2} v_{i,2}^\top.
$$

We next calculate the gradient of $v$, it satisfies

$$
\begin{aligned}
\nabla_{v_i} &= U_i^\top (U_i v_i - w_i) \\
&= U_{i,A}^\top U_{i,A} v_i + U_{i,B}^\top U_{i,B} v_i - U_{i,A}^\top w_{i,A} - U_{i,B}^\top w_{i,B} \\
&= U_{i,A}^\top U_{i,A} v_{i,1} - U_{i,A}^\top w_{i,A} + U_{i,B}^\top U_{i,B} v_{i,2} - U_{i,B}^\top w_{i,B}.
\end{aligned}
$$

The first step follows from Eq. (7), the second step follows from the first two terms of Lemma 4.4. The third step uses the fact that $\mathsf{row}(U_{i,A}) \in \mathsf{V}_{i-1}$, $v_{i,1} \in \mathsf{V}_{i-1}$, $v_{i,2} \in \mathsf{V}_{i-1,\perp}$ and $\mathsf{row}(U_{i,B}) \in \mathsf{V}_{i-1,\perp}$

Hence, we have

$$
\nabla_{v_{i,1}} = U_{i,A}^\top U_{i,A} v_{i,1} - U_{i,A}^\top w_{i,A}
$$

and

$$
\begin{aligned}
\nabla_{v_{i,2}} &= U_{i,B}^\top U_{i,B} v_{i,2} - U_{i,B}^\top w_{i,B} \\
&= (w_{i,B} x_i^\top + U_{i,2})^\top (w_{i,B} x_i^\top + U_{i,2}) v_{i,2} - (w_{i,B} x_i^\top + U_{i,2})^\top w_{i,B} \\
&= x_i \|w_{i,B}\|_2^2 x_i^\top v_{i,2} + U_{i,2}^\top U_{i,2} v_{i,2} - x_i \|w_{i,B}\|_2^2 \\
&= \|w_{i,B}\|_2^2 (x_i^\top v_i - 1) x_i + U_{i,2}^\top U_{i,2} v_{i,2},
\end{aligned}
$$

where the third step holds due to $w_{i,B} \perp \mathsf{column}(U_{i,2})$. We conclude the proof here. $\qquad \square$

Finally, we prove

*Proof of Lemma 4.6.* For any $i \in [k], t \in [T]$, we have

$$
\begin{aligned}
\|U_{i,t} v_{i,t} - w_i\|_2^2 &= \|(U_{i,A} + U_{i,B,t})(v_{i,1,t} + v_{i,2,t}) - w_{i,A} - w_{i,B}\|_2^2 \\
&= \|U_{i,A} v_{i,1,t} + U_{i,B,t} v_{i,2,t} - w_{i,A} - w_{i,B}\|_2^2 \\
&= \|U_{i,A,t} v_{i,1,t} - w_{i,A}\|_2^2 + \|U_{i,B,t} v_{i,2,t} - w_{i,B}\|_2^2 \\
&= \|U_{i,A} v_{i,1,t} - w_{i,A}\|_2^2 + \|(w_{i,B} x_{i,t}^\top + U_{i,2,t}) v_{i,2,t} - w_{i,B}\|_2^2 \\
&= \|U_{i,A} v_{i,1,t} - w_{i,A}\|_2^2 + \|w_{i,B}\|_2^2 (x_{i,t}^\top v_{i,2,t} - 1)^2 + \|U_{i,2,t} v_{i,2,t}\|_2^2.
\end{aligned}
$$

The second step follows from $\mathsf{row}(U_{i,A}) \in \mathsf{V}_{i-1}$, $\mathsf{row}(U_{i,B}) \in \mathsf{V}_{i-1,\perp}$, $v_{i,1,t} \in \mathsf{V}_{i-1}$, $v_{i,2,t} \in \mathsf{V}_{i-1,\perp}$, the third step follows from $U_{i,A} v_{i,1,t} - w_{i,A} \in \mathsf{W}_{i-1}$ and $U_{i,B,t} v_{2,t} - w_{i,B} \in \mathsf{W}_{i-1,\perp}$. The last step follows from $w_{i,B} \perp \mathsf{column}(U_{i,2,t})$. $\qquad \square$

## B.3 Missing proof from Section 4.2.3

In the proof, we write $x = y \pm z$ if $x \in [y - z, y + z]$. For simplicity, we assume $\log(1/\epsilon\nu) \ll k, d$. First, we prove linear convergence for the first loss term.

**Lemma B.1** (Fast learning on existing features). *For any $i \in [k]$ and $t \in [T]$, we have*

$$\|U_{i,A}v_{i,1,t} - w_{i,A}\|_2 \leq \left(1 - \frac{\eta}{4D}\right)^t \|U_{i,A}v_{i,1,0} - w_{i,A}\|_2.$$

*Proof of Lemma B.1.* This follows easily from the standard analysis of gradient descent for least square regressions. For any $t \in [0 : T - 1]$, one has

$$
\begin{aligned}
\|U_{i,A}v_{i,1,t+1} - w_{i,A}\|_2 &= \|U_{i,A}(v_{i,1,t} - \eta(U_{i,A}^\top U_{i,A}v_{i,1,t} - U_{i,A}^\top w_{i,A})) - w_{i,A}\|_2 \\
&= \|(I - \eta U_{i,A}U_{i,A}^\top)(U_{i,A}v_{i,1,t} - w_{i,A})\|_2 \\
&\leq (1 - \frac{\eta}{4D})\|(U_{i,A}v_{i,1,t} - w_{i,A})\|_2.
\end{aligned}
$$

The first step follows from the gradient update formula (see Lemma 4.5), the third step follows from $U_{i,A}v_{i,1,t} - w_{i,A} \in \mathsf{column}(U_{i,A})$, and $2\sqrt{D} \geq \sigma_{\max}(U_{i,A}) \geq \sigma_{\min}(U_{i,A}) \geq \frac{1}{2\sqrt{D}}$ and $\eta < \frac{1}{4D}$. We conclude the proof here. $\square$

We next focus on the second and last loss terms. One can show that $x_{i,t}^\top v_{i,2,t}$ moves to 1 while $\|\|w_{i,B}\|_2 x_{i,t} - v_{i,2,t}\|_2$ remains small in the first $T_1 = O(\frac{D}{\eta} \log \frac{k}{\sigma})$ iterations.

**Lemma B.2.** *With probability at least $1 - O(1/k)$ over the random initialization, there exists $T_1 = O(\frac{D}{\eta} \log \frac{k}{\sigma})$, such that for any $t \leq T_1$, one has (1) $\|\|w_{i,B}\|_2 x_{i,t} - v_{i,2,t}\|_2 \leq O(r\sigma \log(k/\sigma))$; (2) $x_{i,t}^\top v_{i,2,t} < 0.9$ when $t < T_1$ and $0.9 < x_{i,T_1}^\top v_{i,2,T_1} < 1$; (3) $U_{i,2,t}^\top U_{i,2,t} \preceq U_{i,2,0}^\top U_{i,2,0}$.*

*Proof of Lemma B.2.* Recall our goal is to prove

1. $\|\|w_{i,B}\|_2 x_{i,t} - v_{i,2,t}\|_2 \leq O(r\sigma \log(k/\sigma))$,

2. $x_{i,t}^\top v_{i,2,t} < 0.9$ when $t < T_1$ and $0.9 < x_{i,T_1}^\top v_{i,2,T_1} < 1$,

3. $U_{i,2,t}^\top U_{i,2,t} \preceq U_{i,2,0}^\top U_{i,2,0}$.

We inductively prove these three claims. For the base case, we have that

$$
\begin{aligned}
x_{i,0} &= \frac{1}{\|w_{i,B}\|_2^2} w_{i,B}^\top U_{i,B,0} = \frac{1}{\|w_{i,B}\|_2^2} w_{i,B}^\top P_{\mathsf{W}_{i-1,\perp}} U_{i,\text{init}} P_{\mathsf{V}_{i-1,\perp}} = \frac{1}{\|w_{i,B}\|_2^2} w_{i,B}^\top U_{i,\text{init}} P_{\mathsf{V}_{i-1,\perp}} \\
&\approx \frac{\sigma}{\|w_{i,B}\|_2} \cdot \mathsf{rand}(r, 1) P_{\mathsf{V}_{i-1\perp}},
\end{aligned}
\tag{8}
$$

where in the first step we use the fact that $U_{i,B,0} = w_{i,B}x_{i,0}^\top + U_{i,2,0}$, $w_{i,B} \perp \mathsf{column}(U_{i,2,0})$, in the third step, we use $w_{i,B} \in \mathsf{W}_{i-1,\perp}$. The fourth step follows from $\frac{1}{\|w_{i,B}\|_2^2} w_{i,B}^\top U_{i,\text{init}}$ is a random Gaussian vector with variance $\frac{\sigma}{\|w_{i,B}\|_2^2}$. Similarly, we have

$$v_{i,2,0} = \sigma \cdot \mathsf{rand}(r, 1) P_{\mathsf{V}_{i-1,\perp}}. \tag{9}$$

Hence, with probability at least $1 - O(1/k)$, we have

$$\|\|w_{i,B}\|_2 x_{i,0} - v_{i,2,0}\|_2 \leq O(r\sigma \log(k)) \quad \text{and} \quad x_{i,0}^\top v_{i,2,0} < O(\sigma^2 rD \log(k)) \ll 1.$$

We have proved the base case. Now suppose the induction holds up to time $t$, for the $(t+1)$-th iteration, we first go over the first claim. One has

$$\|\|w_{i,B}\|_2 x_{i,t+1} - v_{i,2,t+1}\|_2^2 - \|\|w_{i,B}\|_2 x_{i,t} - v_{i,2,t}\|_2^2$$

$$= \|\|w_{i,B}\|_2(x_{i,t} - \eta v_{i,2,t}(x_{i,t}^\top v_{i,2,t} - 1)) - (v_{i,2,t} - \eta x_{i,t}\|w_{i,B}\|_2^2(x_{i,t}^\top v_{i,2,t} - 1) - \eta U_{i,2,t}^\top U_{i,2,t} v_{i,2,t})\|_2^2$$
$$\quad - \|\|w_{i,B}\|_2 x_{i,t} - v_{i,2,t}\|_2^2$$

$$= \|(\|w_{i,B}\|_2 x_{i,t} - v_{i,2,t}) - \eta v_{i,2,t}\|w_{i,B}\|_2(x_{i,t}^\top v_{i,2,t} - 1) + \eta x_{i,t}\|w_{i,B}\|_2^2(x_{i,t}^\top v_{i,2,t} - 1) + \eta U_{i,2,t}^\top U_{i,2,t} v_{i,2,t}\|_2^2$$
$$\quad - \|\|w_{i,B}\|_2 x_{i,t} - v_{i,2,t}\|_2^2$$

$$= 2\eta\langle\|w_{i,B}\|_2 x_{i,t} - v_{i,2,t}, x_{i,t}\|w_{i,B}\|_2^2(x_{i,t}^\top v_{i,2,t} - 1) - v_{i,2}\|w_{i,B}\|_2(x_{i,t}^\top v_{i,2,t} - 1) + U_{i,2,t}^\top U_{i,2,t} v_{i,2,t}\rangle$$
$$\quad \pm O(\eta^2 D^4)$$

$$= 2\eta\|w_{i,B}\|_2(x_{i,t}^\top v_{i,2,t} - 1)\|\|w_{i,B}\|_2 x_{i,t} - v_{i,2,t}\|_2^2 + \eta\langle\|w_{i,B}\|_2 x_{i,t} - v_{i,2,t}, U_{i,2,t}^\top U_{i,2,t} v_{i,2,t}\rangle$$
$$\quad \pm O(\eta^2 D^4) \tag{10}$$

$$\leq 2\eta\langle\|w_{i,B}\|_2 x_{i,t} - v_{i,2,t}, U_{i,2,t}^\top U_{i,2,t} v_{i,2,t}\rangle \pm O(\eta^2 D^4)$$

$$\leq \widetilde{O}(\eta r \sigma^3 d^2 D) + O(\eta^2 D^4). \tag{11}$$

The first step follows from the gradient update formula (see Lemma 4.5), the third step follows from that

$$\|U_{i,2,t}^\top U_{i,2,t} v_{i,2,t}\|_2 \ll 1, \quad \|\|w_{i,B}\|_2^2(x_{i,t}^\top v_{i,2,t} - 1)x_{i,t}\|_2 \leq O(D^2)$$

and

$$\|\|w_{i,B}\|_2(x_{i,t}^\top v_{i,2,t} - 1)v_{i,2,t}\|_2 \leq O(D^2),$$

which can be derived easily from the induction hypothesis. The fifth step follows from $x_{i,t}^\top v_{i,2,t} < 1$ when $t \leq T_1$. The last step follows from

$$\|\|w_{i,B}\|_2 x_{i,t} - v_{i,2,t}\| \leq \widetilde{O}(r\sigma), \|U_{i,2,t}^\top U_{i,2,t}\| \leq \|U_{i,2,0}^\top U_{i,2,0}\| \leq \widetilde{O}(d^2\sigma^2), \|v_{i,2,t}\|_2 \leq O(D), \tag{12}$$

which can be derived easily from the induction hypothesis. Combining with $\eta \leq \frac{\sigma^2}{D^5}, \sigma \leq \frac{1}{D^2 d^2}$ and the total number of iteration is $T_1 \leq O(\frac{D}{\eta}\log\frac{k}{\sigma})$, one can proved the first claim.

For the second claim, we have that

$$x_{i,t+1}^\top v_{i,2,t+1} - x_{i,t}^\top v_{i,2,t}$$

$$= (x_{i,t} - \eta v_{i,2,t}(x_{i,t}^\top v_{i,2,t} - 1))^\top(v_{i,2,t} - \eta x_{i,t}\|w_{i,B}\|_2^2(x_{i,t}^\top v_{i,2,t} - 1) - \eta U_{i,2,t}^\top U_{i,2,t} v_{i,2,t}) - x_{i,t}^\top v_{i,2,t}$$

$$= -\eta(\|w_{i,B}\|_2^2\|x_{i,t}\|_2^2 + \|v_{i,2,t}\|_2^2)(x_{i,t}^\top v_{i,2,t} - 1) - \eta x_{i,t}^\top U_{i,2,t}^\top U_{i,2,t} v_{i,2,t} \pm O(\eta^2 D^3) \tag{13}$$

$$\geq \frac{1}{2}\eta(\|w_{i,B}\|_2^2\|x_{i,t}\|_2^2 + \|v_{i,2,t}\|_2^2)(x_{i,t}^\top v_{i,2,t} - 1) - O(\eta^2 D^3)$$

$$\geq \frac{1}{20}\eta(\|w_{i,B}\|_2^2\|x_{i,t}\|_2^2 + \|v_{i,2,t}\|_2^2) - O(\eta^2 D^3). \tag{14}$$

The first step follows from the gradient update formula (see Lemma 4.5), the second step holds since

$$\|v_{i,2,t}(x_{i,t}^\top v_{i,2,t} - 1))\|_2 \leq O(D), \quad \|\|w_{i,B}\|_2^2(x_{i,t}^\top v_{i,2,t} - 1)x_{i,t}\|_2 \leq O(D^2) \quad \text{and} \quad \|U_{i,2,t}^\top U_{i,2,t} v_{i,2,t}\|_2 \ll 1.$$

Again, these inequalities can be derived easily from the inductive hypothesis. The third step holds since $U_{i,2,t}^\top U_{i,2,t} \preceq U_{i,2,t}^\top U_{i,2,t} \preceq \widetilde{O}(d^2\sigma^2) \cdot I$, and therefore,

$$|x_{i,t}^\top U_{i,2,t}^\top U_{i,2,t} v_{2,t}| \leq \widetilde{O}(d^2\sigma^2) \cdot \|x_{i,t}\|_2\|v_{i,2,t}\|_2 \ll |(\|w_{i,B}\|_2^2\|x_{i,t}\|_2^2 + \|v_{i,2,t}\|_2^2)(x_{i,t}^\top v_{i,2,t} - 1)|.$$

The last step uses the fact that $x_{i,t}^\top v_{2,t} < 0.9$ when $t < T_1$.

We next bound the RHS of Eq. (14) and prove it can not be too small. We focus on $\|\|w_{i,B}\|_2 x_{i,t+1} + v_{i,2,t+1}\|_2$ and prove it monotonically increasing. In particular, at initialization, with probability at least $1 - O(1/k)$, due to anti-concentration of Gaussian, we have

$$\|\|w_{i,B}\|x_{i,0} + v_{i,2,0}\|_2 \approx \sigma\|\mathsf{rand}(r,1)P_{\mathsf{V}_{i-1,\perp}}\|_2 \geq \sigma/k. \tag{15}$$

Furthermore, we have

$$\left\|\|w_{i-1,B}\|_2 x_{i,t+1} + v_{i,2,t+1}\right\|_2^2$$
$$= \left\|\|w_{i-1,B}\|_2(x_{i,t} - \eta v_{i,2,t}(x_{i,t}^\top v_{i,2,t} - 1)) + (v_{i,2,t} - \eta x_{i,t}\|w_{i,B}\|_2^2(x_{i,t}^\top v_{i,2,t} - 1) - \eta U_{i,2,t}^\top U_{i,2,t} v_{i,2,t})\right\|_2^2$$
$$= \left\|\|w_{i,B}\|_2 x_{i,t} + v_{i,2,t}\right\|_2^2 + 2\eta\|w_{i,B}\|_2(1 - x_{i,t}^\top v_{i,2,t})\left\|\|w_{i,B}\|_2 x_{i,t} + v_{i,2,t}\right\|_2^2$$
$$\quad + \eta\langle\|w_{i,B}\|_2 x_{i,t} + v_{i,2,t}, U_{i,2,t}^\top U_{i,2,t} v_{i,2,t}\rangle \pm O(\eta^2 D^4)$$
$$\geq (1 + \frac{1}{20}\eta\|w_{i,B}\|_2)\|w_{i,B} x_{i,t} + v_{i,2,t}\|_2^2, \tag{16}$$

where the first step holds due to the gradient update formula (see Lemma 4.5), the second step holds due to Eq. (12). The last step holds since

$$\|U_{i,2,t}^\top U_{i,2,t} v_{i,2,t}\|_2 \leq \widetilde{O}(d^2\sigma^2 D) \ll \frac{\sigma}{40kD} \leq \frac{1}{40}\|w_{i,B}\|_2 \cdot \left\|\|w_{i,B}\|_2 x_{i,0} + v_{i,2,0}\right\|_2$$
$$\leq \frac{1}{40}\|w_{i,B}\|_2 \cdot \left\|\|w_{i,B}\|_2 x_{i,t} + v_{i,2,t}\right\|_2$$

and

$$O(\eta D^4) \ll \frac{\sigma^2}{40k^2 D} \leq \frac{1}{40}\|w_{i,B}\|_2 \cdot \left\|\|w_{i,B}\|_2 x_{i,0} + v_{i,2,0}\right\|_2^2 \leq \frac{1}{40}\|w_{i,B}\|_2 \cdot \left\|\|w_{i,B}\|_2 x_{i,t} + v_{i,2,t}\right\|_2^2.$$

Hence, we conclude that $\left\|\|w_{i,B}\|_2 x_t + v_{2,t}\right\|_2$ is monotonically increasing, and in particular,

$$\left\|\|w_{i,B}\|_2 x_{i,t} + v_{i,2,t}\right\|_2^2 \geq \left\|\|w_{i,B}\|_2 x_{i,0} + v_{i,2,0}\right\|_2^2 = \Omega(\sigma^2/k^2) \qquad \forall t \in [T_1]$$
$$\left\|\|w_{i,B}\|_2 x_{i,t} + v_{i,2,t}\right\|_2^2 \geq \Omega(1) \qquad\qquad t \geq O(\frac{D}{\eta}\log\frac{k}{\sigma})$$

The second inequality follows from Eq. (16). Plugging into Eq. (14), one has

$$x_{i,t+1}^\top v_{i,2,t+1} - x_{i,t}^\top v_{i,2,t} \geq \frac{1}{20}\eta(\|w_{i,B}\|_2^2\|x_{i,t}\|_2^2 + \|v_{i,2,t}\|_2^2) - O(\eta^2 D^3)$$
$$\geq \frac{1}{40}\eta(\left\|\|w_{i,B}\|_2 x_{i,t} + v_{i,2,t}\right\|_2^2) - O(\eta^2 D^3)$$
$$\geq \begin{cases} 0 & t \in [T] \\ \Omega(\eta) & t \geq O(\frac{D}{\eta}\log\frac{k}{\sigma}) \end{cases}$$

Hence, after at most $T_1 \leq O(\frac{D}{\eta}\log\frac{k}{\sigma})$ iterations, we have $0.9 \leq x_{i,T_1}^\top v_{i,2,T_1} < 1$. It would not exceed 0.9 too much since by Eq. (13), the change per iteration is at most

$$|x_{i,t+1}^\top v_{i,2,t+1} - x_{i,t}^\top v_{i,2,t}| \lesssim \eta(\|w_{i,B}\|_2^2\|x_{i,t}\|_2^2 + \|v_{i,2,t}\|_2^2)$$
$$\leq \eta(\left\|\|w_{i,B}\|_2 x_t - v_{i,2,t}\right\|_2^2 + 2\|w_{i,B}\|_2 x_{i,t}^\top v_{i,2,t}) \leq 4\eta D \ll 1 \tag{17}$$

For the third claim, we have

$$U_{i,2,t+1}^\top U_{i,2,t+1} = (U_{i,2,t} - \eta U_{i,2,t} v_{i,2,t} v_{i,2,t}^\top)^\top(U_{i,2,t} - \eta U_{i,2,t} v_{i,2,t} v_{i,2,t}^\top) \preceq U_{i,2,t}^\top U_{i,2,t}.$$

The last step holds since $(I - v_{i,2,t} v_{i,2,t}^\top)$ is a PSD matrix and $(I - v_{i,2,t} v_{i,2,t}^\top) \preceq I$. We have proved all three claims. $\qquad\square$

A linear convergence of the second and the last loss terms can be shown, after the first $T_1$ iterations.

**Lemma B.3.** *Let $T_2 = O(\frac{D}{\eta}\log(\frac{kdD}{\epsilon\nu}))$. After $T = T_1 + T_2$ iterations, we have (1) $|x_{i,T}^\top v_{i,2,T} - 1| \leq \epsilon\nu/kdD$; (2) $\|U_{i,2,T}^\top U_{i,2,T} v_{i,2,T}\|_2 \leq \epsilon\nu$.*

*Proof of Lemma B.3.* For the $t$-th iteration ($t \in [T_1 : T_2]$), we prove the following claims inductively.

1. $|x_{i,t}^\top v_{i,2,t} - 1| \leq \frac{1}{2}(1 - \frac{\eta}{4D})^{t-T_1}$,

2. $\|U_{i,t}v_{i,t}\|_2 \le (1 - \frac{\eta}{4D})^{t-T_1}$,

3. $\|\|w_{i,B}\|_2 x_{i,t} - v_{i,2,t}\|_2 \le \widetilde{O}(r\sigma)$.

The inductive base ($t = T_1$) holds trivially. Assuming the hypothesis holds up to time $t$, we start from the first claim. We have that

$$(1 - x_{i,t+1}^\top v_{i,2,t+1}) - (1 - x_{i,t}^\top v_{i,2,t})$$
$$= -(x_{i,t} - \eta v_{i,2,t}(x_t^\top v_{i,2,t} - 1))^\top (v_{i,t} - \eta x_{i,t}\|w_{i,B}\|_2^2(x_{i,t}^\top v_{i,2,t} - 1) - \eta U_{i,2,t}^\top U_{i,2,t}v_{i,2,t}) + x_{i,t}^\top v_{i,2,t}$$
$$= \eta(\|w_{i,B}\|_2^2\|x_{i,t}\|_2^2 + \|v_{i,2,t}\|_2^2)(x_{i,t}^\top v_{i,2,t} - 1) + \eta x_{i,t}^\top U_{i,2,t}^\top U_{i,2,t}v_{i,2} \pm O(\eta^2 D^3 |x_{i,t}^\top v_{i,2,t} - 1|).$$

The first step follows from the gradient update formula (see Lemma 4.5), the second step follows from

$$\|w_{i,B}\|_2^2\|x_{i,t}\| \le O(D^2), \quad \|v_{i,2,t}\|_2 \le D \quad \text{and} \quad \|U_{i,2,t}v_{i,2,t}\|_2 \ll 1.$$

Since

$$\|w_{i,B}\|_2^2\|x_{i,t}\|_2^2 + \|v_{i,2,t}\|_2^2 = \|\|w_{i,B}\|x_{i,t} - v_{i,2,t}\|_2^2 + 2\langle\|w_{i,B}\|_2 x_{i,t}, v_{i,2,t}\rangle \ge \frac{1}{D}$$

holds due to our inductive hypothesis, we further have that

$$|1 - x_{i,t+1}^\top v_{i,2,t+1}| \le (1 - \frac{\eta}{D})|1 - x_{i,t}^\top v_{i,2,t}| + \eta|x_{i,t}^\top U_{i,2,t}^\top U_{i,2,t}v_{i,2,t}| \pm O(\eta^2 D^3 |x_{i,t}^\top v_{i,2,t} - 1|).$$
$$(18)$$

**Case 1.** Suppose $\frac{1}{2}(1 - \frac{\eta}{4D})^{t+2-T_1} \le |x_{i,t}^\top v_{i,2,t} - 1| \le \frac{1}{2}(1 - \frac{\eta}{4D})^{t-T_1}$, then we have

$$|1 - x_{i,t+1}^\top v_{i,2,t+1}| \le (1 - \frac{\eta}{4D})|1 - x_{i,t}^\top v_{i,2,t}| \le \frac{1}{2}(1 - \frac{\eta}{4D})^{t+1-T_1}.$$

This holds due to Eq. (18), $\eta D^3 \ll \frac{1}{4D}$ and

$$|x_{i,t}^\top U_{i,2,t}^\top U_{i,2,t}v_{i,2,t}| \le \|x_{i,t}^\top U_{i,2,t}^\top\|_2\|U_{i,2,t}v_{i,2,t}\|_2 \le \widetilde{O}(Dd\sigma) \cdot 2|x_{i,t}^\top v_{i,2,t} - 1| \le \frac{1}{4D}|x_{i,t}^\top v_{i,2,t} - 1|,$$

where the second step holds due to the induction hypothesis.

**Case 2.** Suppose $|x_{i,t}^\top v_{i,2,t} - 1| \le \frac{1}{2}(1 - \frac{\eta}{4D})^{t+2-T_1}$, then we have

$$\eta|x_{i,t}^\top U_{i,2,t}^\top U_{i,2,t}v_2| \pm O(\eta^2|x_{i,t}^\top v_{i,2,t} - 1|D^3) \le \eta \cdot \widetilde{O}(Dd\sigma) \cdot (1 - \frac{\eta}{4D})^{t-T_1} + O(\eta^2 D^3) \cdot (1 - \frac{\eta}{4D})^{t-T_1}$$
$$\le \frac{1}{2}(1 - \frac{\eta}{4D})^{t+1-T_1} \cdot \frac{\eta}{4D},$$

where the first step holds due to induction hypothesis and

$$|x_{i,t}^\top U_{i,2,t}^\top U_{i,2,t}v_{i,2}| \le \|x_{i,t}^\top U_{i,2,t}^\top\|_2\|U_{i,2,t}v_{i,2,t}\|_2 \le \widetilde{O}(Dd\sigma) \cdot (1 - \frac{\eta}{4D})^{t-T_1}.$$

Therefore

$$|1 - x_{i,t+1}^\top v_{i,2,t+1}| \le \frac{1}{2}(1 - \frac{\eta}{4D})^{t+2-T_1} + \frac{1}{2}(1 - \frac{\eta}{4D})^{t+1-T_1} \cdot \frac{\eta}{4D} = \frac{1}{2}(1 - \frac{\eta}{4D})^{t+1-T_1}.$$

Next, we prove the second claim. We have

$$\|U_{i,2,t+1}v_{i,2,t+1}\|_2$$
$$= \|(U_{i,2,t} - \eta U_{i,2,t}v_{i,2,t}v_{i,2,t}^\top)(v_{i,2,t} - \eta x_{i,t}\|w_{i,B}\|_2^2(x_{i,t}^\top v_{i,2,t} - 1) - \eta U_{i,2,t}^\top U_{i,2,t}v_{i,2,t})\|_2$$
$$\le \|U_{i,2,t}v_{i,2,t}(1 - \eta v_{i,2,t}^\top v_{i,2,t}) - \eta U_{i,2,t}U_{i,2,t}^\top U_{i,2,t}v_{i,2,t}\|_2 + \eta\|w_{i,B}\|_2^2|x_{i,t}^\top v_{i,2,t} - 1|\|U_{i,2,t}x_{i,t}\|_2$$
$$\quad \pm O(\eta^2 D^3) \cdot \|U_{i,2,t}v_{i,2,t}\|_2$$
$$\le (1 - 5\eta\|v_{i,2,t}\|_2^2/6)\|U_{i,2,t}v_{i,2,t}\|_2 + \eta\|w_{i,B}\|_2^2|x_{i,t}^\top v_{i,2,t} - 1|\|U_{i,2,t}x_{i,t}\|_2$$
$$\le (1 - \frac{\eta}{3D})\|U_{i,2,t}v_{i,2,t}\|_2 + \eta\|w_{i,B}\|_2^2|x_{i,t}^\top v_{i,2,t} - 1|\|U_{i,2,t}x_{i,t}\|_2, \quad (19)$$

where the first step follows from the gradient update rule (Lemma 4.5), the second step holds due to triangle inequality and

$$\|v_{i,2,t}\|_2 \le O(D), \quad |x_{i,t}^\top v_{i,2,t} - 1|\|w_{i,B}\|_2^2\|x_{i,t}\|_2 \le O(D^2) \quad \text{and} \quad \|U_{i,2,t}^\top U_{i,2,t}v_{i,2,t}\| \ll 1,$$

the third step holds due to $\eta D^3 \le \|v_{i,2,t}\|_2^2/6$ and the last step holds since

$$\begin{aligned}
\|v_{i,2,t}\|_2^2 &= v_{i,2,t}^\top(\|w_{i,B}\|_2 x_{i,t} + v_{i,2,t} - \|w_{i,B}\|_2 x_{i,t}) \\
&\ge \|w_{i,B}\|_2 x_{i,2,t}^\top v_{i,2,t} - \|v_{i,2,t}\|_2\|w_{i,B}\|_2 x_{i,t} - v_{i,2,t}\|_2 \\
&\ge \frac{1}{2D} - \widetilde{O}(Dr\sigma) \ge \frac{2}{5D}.
\end{aligned}$$

**Case 1.** Suppose $(1 - \frac{\eta}{4D})^{t+2-T_1} \le \|U_{i,2,t}v_{2,t}\|_2 \le (1 - \frac{\eta}{4D})^{t-T_1}$, then

$$\begin{aligned}
\|U_{i,2,t+1}v_{i,2,t+1}\|_2 &\le (1 - \frac{\eta}{3D})\|U_{i,2,t}v_{i,2,t}\|_2 + \eta\|w_{i,B}\|_2^2|x_{i,t}^\top v_{i,2,t} - 1|\|U_{i,2,t}x_{i,t}\|_2 \\
&\le (1 - \frac{\eta}{4D})\|U_{i,2,t}v_{i,2,t}\|_2 \le (1 - \frac{\eta}{4D})^{t+1-T_1},
\end{aligned}$$

where the first step comes from Eq. (19), the second step comes from

$$\begin{aligned}
\eta\|w_{i,B}\|_2^2|x_{i,t}^\top v_{i,2,t} - 1|\|U_{i,2,t}x_{i,t}\|_2 &\le \eta D^2 \cdot \frac{1}{2}(1 - \frac{\eta}{4D})^{t-T_1} \cdot \widetilde{O}(Dd\sigma) \\
&\le \frac{\eta}{12D}(1 - \frac{\eta}{4D})^{t+2-T_1} \le \frac{\eta}{12D}\|U_{i,2,t}v_{i,2,t}\|_2.
\end{aligned}$$

**Case 2.** Suppose $\|U_t v_{2,t}\|_2 \le (1 - \frac{\eta}{4D})^{t+2-T_1}$, then

$$\begin{aligned}
\|U_{i,2,t+1}v_{i,2,t+1}\|_2 &\le \|U_{i,2,t}v_{i,2,t}\|_2 + \eta\|w_{i,B}\|_2^2 \cdot |x_{i,t}^\top v_{i,2,t} - 1| \cdot \|U_{i,2,t}x_{i,t}\|_2 \\
&\le (1 - \frac{\eta}{4D})^{t+2-T_1} + \frac{1}{2}\eta D^2(1 - \frac{\eta}{4D})^{t-T_1} \cdot \widetilde{O}(Dd\sigma) \\
&\le (1 - \frac{\eta}{4D})^{t+2-T_1} + (1 - \frac{\eta}{4D})^{t+1-T_1} \cdot \frac{\eta}{4D} \\
&= (1 - \frac{\eta}{4D})^{t+1-T_1},
\end{aligned}$$

where the first step comes from Eq. (19), the second step follows from the induction hypothesis and $\|U_{i,2,t}x_{i,t}\|_2^2 \le \widetilde{O}(Dd\sigma)$. We have proved the second claim.

Now we move to the third claim. One has

$$\begin{aligned}
&\|\|w_{i,B}\|_2 x_{i,t+1} - v_{i,2,t+1}\|_2^2 - \|\|w_{i,B}\|_2 x_{i,t} - v_{i,2,t}\|_2^2 \\
&= 2\eta(x_{i,t}^\top v_{i,2,t} - 1)\|w_{i,B}\|_2\|\|w_{i,B}\|_2 x_{i,t} - v_{i,2,t}\|_2^2 + \eta\langle\|w_{i,B}\|_2 x_{i,t} - v_{i,2,t}, U_{i,2,t}^\top U_{i,2,t}v_{i,2,t}\rangle \pm O(\eta^2 D^4) \\
&\lesssim 2\eta \cdot d^2 D^3\sigma^2 \cdot D \cdot (r\sigma)^2 + \eta \cdot r\sigma \cdot d^2\sigma^2 D + \eta^2 D^4 \\
&\lesssim \eta D d^2 r\sigma^3.
\end{aligned}$$

The first step comes from Eq. (10), the second step follows from

$$\|w_{i,B}\|_2 \le D, \quad \|\|w_{i,B}\|_2 x_{i,t} - v_{i,2,t}\|_2 \le \widetilde{O}(r\sigma), \quad \|U_{i,2,t}^\top U_{i,2,t}v_{i,2,t}\|_2 \le \widetilde{O}(d^2\sigma^2 D)$$

and

$$x_{i,t}^\top v_{i,2,t} - 1 \le \widetilde{O}(d^2 D^3\sigma^2).$$

Here the last term holds since (i) $|x_{i,\tau+1}^\top v_{i,2,\tau+1} - x_{i,\tau}^\top v_{i,2,\tau}| \le O(\eta D)$, i.e., the step size is at most $\eta D$ (see Eq. (13) (17)); (ii) $x_{i,T_1}^\top v_{i,2,T_1} < 1$ and (iii) $|x_{i,\tau+1}^\top v_{i,2,\tau+1} - 1| < |x_{i,\tau}^\top v_{i,2,\tau} - 1|$ whenever

$$\eta\|x_{i,t}^\top U_{i,2,t}^\top U_{i,2,t}v_2\|_2 \le \eta \cdot \widetilde{O}(d^2 D^2\sigma^2) \lesssim \frac{\eta}{2D}|x_{i,\tau}^\top v_{i,2,\tau} - 1| \quad \Rightarrow \quad |x_{i,\tau}^\top v_{i,2,\tau} - 1| \gtrsim d^2 D^3\sigma^2.$$

That is, combining (i) (ii), we know that the first time $x_{i,\tau}^\top v_{i,\tau}$ being greater 1 must obey $x_{i,\tau}^\top v_{i,\tau} < 1 + O(\eta D)$, (iii) implies that whenever $x_{i,\tau+1}^\top v_{i,2,\tau+1} - 1 \gtrsim d^2 D^3\sigma^2$, it value should decrease, hence we conclude

$$x_{i,T_1}^\top v_{i,2,T_1} - 1 \lesssim \eta D + d^2 D^3\sigma^2 \lesssim d^2 D^3\sigma^2.$$

Taking a telescopic summation, one has

$$\|\|w_{i,B}\|_2 x_{i,t} - v_{i,2,t}\|_2^2 - \|\|w_{i,B}\|_2 x_{i,T_1} - v_{i,2,T_1}\|_2^2 \le (t - T_1) \cdot O(\eta D d^2 r^2 \sigma^3)$$
$$\le \widetilde{O}(D^2 d^2 r \sigma^3) \le r^2 \sigma^2.$$

This concludes the third claim. We conclude the proof here. $\qquad\square$

Combining Lemma 4.6, Lemma B.1 – B.3, one can conclude the proof of Lemma 4.7.

## B.4  Missing proof from Section 4.2.4

*Proof of Theorem 2.11.* Due to the reduction established in Section 4.2.1, it suffices to prove Eq. (3) and Eq. (4). For each environment $i$ ($i \in [k]$), we inductively prove

1. DPGrad achieves good accuracy on the current environment, i.e., $\|U_{i,T} v_i - w_i\|_2 \le \epsilon\nu$;

2. The feature matrix $U_i$ remains well conditioned, i.e. $\frac{1}{2\sqrt{D}} \le \sigma_{\min}(U_{i,\text{end}}) \le \sigma_{\max}(U_{i,\text{end}}) \le 2\sqrt{D}$.

3. The algorithm does not suffer from catastrophic forgetting, i.e., $\|U_{i,t} v_j - w_i\|_2 \le \epsilon$ for any $j < i$ and $t \in [T]$;

The base case ($i = 0$) holds trivially as at the beginning of CL, we have $\mathsf{W}, \mathsf{V} = \emptyset$ and $U = 0$. Suppose the induction holds up to the $(i-1)$-th environment, we focus on the second and last claim, as the first claim holds directly due to Lemma 4.7.

For the second claim, we have already proved $\|U_{i,T} v_i - w_i\|_2 \le \epsilon\nu$, this indicates that each coordinate of $U_{i,T} v_i - w_i$ is less than $\nu/2$. Since we assume each coordinate of $w_i$ is a multiple of $\nu$, therefore, we have $\widehat{w}_i = \mathsf{Round}_\nu(U_{i,T} v_i) = w_i$. That is, we exact recover $w_i$. We divide into two cases.

**Case 1.** If $\|w_{i,B}\|_2 = 0$, i.e., $w_i \in \mathsf{W}$, then $\|P_{\mathsf{W}_\perp} \widehat{w}_i\|_2 = \|P_{\mathsf{W}_\perp} w_i\|_2 = 0$, Therefore, we do not update $\mathsf{W}$ and $\mathsf{V}$, and

$$U_{i,\text{end}} = P_{\mathsf{W}} U_{i,T} P_{\mathsf{V}} = P_{\mathsf{W}}(U_{i,A,0} + U_{i,B,T}) P_{\mathsf{V}} = P_{\mathsf{W}} U_{i,A,0} P_{\mathsf{V}} = U_{i-1,\text{end}},$$

where the second and the third step holds to Lemma 4.4 and the last step just holds due to definition. Hence $U_i$ continues to be well-conditioned (since it does not change).

**Case 2.** If $\|w_{i,B}\|_2 \in [1/D, D]$, then $\|P_{\mathsf{W}_\perp} \widehat{w}_i\|_2 = \|P_{\mathsf{W}_\perp} w_i\|_2 = \|w_{i,B}\| \ge 1/D$. Hence, we augment $\mathsf{W}_i = \mathsf{W}_{i-1} \cup \{w_i\}$ and $\mathsf{V}_i = \mathsf{V}_{i-1} \cup \{v_i\}$ and have

$$\begin{aligned}
U_{i,\text{end}} &= P_{\mathsf{W}} U_{i,T} P_{\mathsf{V}} = P_{\mathsf{W}}(U_{i,A,0} + U_{i,B,T}) P_{\mathsf{V}} \\
&= U_{i,A,0} + (\frac{1}{\|w_{i,B}\|_2^2} w_{i,B} w_{i,B}^\top) U_{i,B,T} (\frac{1}{\|v_{i,2,T}\|_2^2} v_{i,2,T} v_{i,2,T}^\top) \\
&= U_{i,A,0} + (\frac{1}{\|w_B\|_2^2} w_B w_B^\top)(w_B x_{i,T}^\top + U_{i,2,T})(\frac{1}{\|v_{i,2,T}\|_2^2} v_{i,2,T} v_{i,2,T}^\top) \\
&= U_{i,A,0} + w_B v_{i,2,T}^\top \frac{x_{i,T}^\top v_{i,2,T}}{\|v_{i,2,T}\|_2^2} \\
&= U_{i,A,0} + (1 \pm o(\epsilon/D)) \frac{1}{\|v_{i,2,T}\|_2^2} w_B v_{i,2,T}^\top.
\end{aligned} \tag{20}$$

The third step holds since $\mathsf{row}(U_{i,A,0}) \in \mathsf{V}$, $\mathsf{column}(U_{i,A,0}) \in \mathsf{W}$, $\mathsf{column}(U_{i,B,T}) \cap \mathsf{W} = w_{i,B}$, $\mathsf{row}(U_{i,B,T}) \cap \mathsf{V} = v_{i,2,T}$ (see Lemma 4.4), the later two imply the projection operation essentially boils to projection on $w_{i,B}$ and $v_{i,2,T}$. The fifth step follows from $\mathsf{column}(U_{i,2,T}) \perp w_B$ (see Lemma 4.4), the sixth step follows from $x_{i,T}^\top v_{i,T} = 1 \pm o(\epsilon/D)$ (see Lemma B.3). To bound the condition number, it suffices to note that $w_{i,B} \perp \mathsf{W}_{i-1}$, $v_{i,2,T} \perp \mathsf{V}_{i-1}$ (see Lemma 4.4), and therefore, $w_B \perp \mathsf{column}(U_{i,A,0})$, $v_{i,2,T} \perp \mathsf{row}(U_{i,A,0})$ (i.e., we add an orthogonal basis) and

$$(1 \pm o(\epsilon/D)) \frac{1}{\|v_{i,2,T}\|_2^2} \|w_B\|_2 \|v_{i,2,T}^\top\|_2 = (1 \pm o(\epsilon/D)) \frac{\|w_B\|_2}{\|v_{i,2,T}\|_2} = (1 + o(1)) \sqrt{\|w_B\|_2} \in \left[\frac{1}{2\sqrt{D}}, \frac{\sqrt{D}}{2}\right]$$

where the last step is derived from $x_{i,t}^\top v_{i,2,T} \approx 1 + o(1)$ and $\|\|w_B\|_2 x_{i,t} - v_{i,2,t}\|_2 \approx 1 \pm o(1/D^3)$. We have proved the second claim.

For the last claim, fix an index $j < i$, we prove the accuracy of $j$-th environment would not drop significantly and remain good. Note by inductive hypothesis, we already have $\|U_{j,T} v_j - w_j\|_2 \leq \epsilon\nu/kd$ before the final projection step of $j$-th environment. After the projection step, one has

$$\|U_{j,\text{end}} v_j - w_j\|_2 = \|P_\mathsf{W} U_{j,T} P_\mathsf{V} v_j - w_j\|_2 = \|P_\mathsf{W}(U_{j,A,T} + w_{j,B} x_{j,T}^\top + U_{j,2,T}) P_\mathsf{V} v_j - w_j\|_2$$

We divide into two cases.

**Case 1.** Suppose $\|w_{j,B}\|_2 = 0$. We have $\mathsf{W}_j = \mathsf{W}_{j-1}, \mathsf{V}_j = \mathsf{V}_{j-1}$ and

$$\|U_{j,\text{end}} v_j - w_j\|_2 = \|P_{\mathsf{W}_j}(U_{j,A,T} + U_{j,2,T}) P_{\mathsf{V}_j} v_j - w_j\|_2 = \|U_{j,A,T} v_j - w_j\|_2$$
$$\leq \|U_{j,A,T} v_{j,1,T} - w_j\|_2 \leq \epsilon\nu.$$

The second step follows from $\text{column}(U_{j,A,T}) \in \mathsf{W}_j$, $\text{row}(U_{j,A,T}) \in \mathsf{V}_k$ and $\text{row}(U_{j,2,T}) \in \mathsf{V}_{j,\perp}$ (see Lemma 4.4), the third step follows from $\text{row}(U_{j,A,T}) \in \mathsf{V}_{i-1}$. Hence, we have that the error remains small after the projection.

During the $i$-th environment, for any $t \in [T]$, we decompose $U_{i,t} = U_{j,\text{end}} + \widehat{U}_{i,t}$. We have

$$\|U_{i,t} v_j - w_j\|_2 = \|(U_{j,\text{end}} + \widehat{U}_{i,t}) v_j - w_j\|_2$$
$$\leq \|U_{j,\text{end}} v_j - w_j\|_2 + \|\widehat{U}_{i,t} v_j\|_2$$
$$= \|U_{j,\text{end}} v_j - w_j\|_2 + \|\widehat{U}_{i,t} v_{j,2,T}\|_2$$
$$\leq \epsilon\nu + \widetilde{O}(\sqrt{D} \cdot r\sigma)$$
$$\leq \epsilon.$$

The third step holds due to the fact that $\text{row}(\widehat{U}_{i,t}) \in V_{j,\perp}$, the fourth step holds due to (1) $\|v_{j,2,t}\|_2$ is non-decreasing during the $j$-th environment (see the gradient update formula in Lemma 4.5) and therefore $\|v_{j,2,T}\|_2 \leq \|v_{j,2,0}\|_2 \leq \widetilde{O}(r\sigma)$ w.h.p.; (2) the spectral norm $\|\widehat{U}_{i,t}\| \leq O(\sqrt{D})$, since

$$\|\widehat{U}_{i,t}\| \leq \|U_{i,t}\| + \|U_{j,\text{end}}\|_2 \leq \|U_{i,A}\| + \|w_{i,B} x_{i,t}^\top + U_{i,2,T}\| + \|U_{j,\text{end}}\|_2$$
$$\leq 2\sqrt{D} + 2\sqrt{D} + 2\sqrt{D} = O(\sqrt{D}).$$

Here the first step and the second step hold due to triangle inequality, the second step holds due to the inductive hypothesis and $\|w_{i,B} x_{i,t}^\top + U_{i,2,T}\| \leq 2\sqrt{D}$. We finished the proof of the first case.

**Case 2.** Suppose $\|w_{j,B}\|_2 \in [1/D, D]$. Then we augment $\mathsf{W}_j = \mathsf{W}_{j-1} \cup \{w_j\}$ and $\mathsf{V}_j = \mathsf{V}_{j-1} \cup \{v_j\}$. We first prove the loss remains small after the final projection step of $j$-th environment. In particular, we have

$$\|U_{j,\text{end}} v_j - w_j\|_2 = \|(U_{j,A,0} + (1 \pm o(\epsilon/D)) \frac{1}{\|v_{j,2,T}\|_2^2} w_B v_{j,2,T}^\top)(v_{j,1,T} + v_{j,2,T}) - w_{j,A} - w_{j,B}\|_2$$
$$= \|(U_{j,A,0} v_{j,1,T} - w_{j,A}) + (1 \pm o(\epsilon/D)) \frac{1}{\|v_{j,2,T}\|_2^2} w_{j,B} v_{j,2,T}^\top v_{j,2,T} - w_{j,B}\|_2$$
$$\leq \|(U_{j,A,0} v_{j,1,T} - w_{j,A})\|_2 + o(\epsilon/D)\|w_{j,B}\|_2$$
$$\leq \epsilon\nu + o(\epsilon) \leq \epsilon.$$

The first step holds due to Eq. (20), the third step holds due to triangle inequality, the fourth step holds due to the inductive hypothesis and $\|w_{j,B}\|_2 \leq D$.

During the $i$-th environment, since the update is performed in the orthogonal space, we expect $U v_j$ does not change. Formally, let $U_{i,t} = U_{j,\text{end}} + \widehat{U}_{i,t}$, where $\text{column}(\widehat{U}_{i,t}) \perp \mathsf{W}_j$ and $\text{row}(\widehat{U}_{i,t}) \perp \mathsf{V}_j$, then

$$U_{i,t} v_j = (U_{j,\text{end}} + \widehat{U}_{i,t}) v_j = U_{j,\text{end}} v_j,$$

Hence $\|U_{i,t} v_j - w_j\| \leq \epsilon$ continues to hold. We conclude the proof here.

$\square$

## C  Missing proof from Section 5

*Proof of Theorem 2.12.* We take $k = 2, n = 3, d = 2$. For both environments, we assume the input data are drawn uniformly at random from $\mathcal{B}_3(0, 1)$, where $\mathcal{B}_3(0, 1)$ denotes the unit ball in $\mathbb{R}^3$ centered at origin. The hypothesis class $\mathcal{H}$ consists of all two-layer convolutional neural network with a single kernel of size 2 and the quadratic activation function. That is, the representation function is parameterized by $w \in \mathbb{R}^2$ and takes the form of $R_w(x) = (\langle w, x_{1:2}\rangle^2, \langle w, x_{2:3}\rangle^2) \in \mathbb{R}^2$, where $x \in \mathbb{R}^3$, $x_{i:j} \in \mathbb{R}^{j-i+1}$ is a vector consists of the $i$-th entry to the $j$-th entry of $x$.

The hard sequence of environments are drawn from the following distribution.

- The objective function $f_1$ of the first environment is $f_1(x) = x_2^2$

- The objective function $f_2$ of the second environment equals $f_2(x) = x_3^2$ with probability $1/2$, and equals $f_2(x) = x_1^2$ with probability $1/2$.

First, the continual learning task is realizable: (1) if $f_2(x) = x_3^2$, then one can take $w = (0, 1)$ and $v_1 = (1, 0), v_2 = (0, 1)$; (2) if $f_2(x) = x_1^2$, then one can take $w = (1, 0), v_1 = (0, 1), v_2 = (1, 0)$.

We then prove no (proper) continual learning algorithm can guarantee to achieve less than $1/1000$-error on both environments with probability at least $1/2$. Suppose the algorithm takes $v_1 = (v_{1,1}, v_{1,2})$ for the first environment. Due to symmetry, one can assume $|v_{1,1}| \geq |v_{1,2}|$. With probability $1/2$, the objective function of the second environment is $f_2(x) = x_1^2$. Let $v_2 = (v_{2,1}, v_{2,2})$ be the linear prompt and $w = (w_1, w_2)$ be the parameter of neural network. We prove by contradiction and assume

$$\mathbb{E}_{x \sim \mathcal{B}_3(0,1)}[|\langle v_1, R_w(x)\rangle - x_2^2|^2] \leq 1/1000 \text{ and } \mathbb{E}_{x \sim \mathcal{B}_3(0,1)}[|\langle v_2, R_w(x)\rangle - x_1^2|^2] \leq 1/1000.$$

Let $\Pi_n^d$ be the space of all polynomial of degree at most $d$ in $n$ variables. By Lemma C.1, notice that $\langle v_1, R_w(x)\rangle, \langle v_2, R_w(x)\rangle \in \Pi_3^2$, we must have that their coefficients match well with $x_2^2$ and $x_1^2$ respectively (in the sense that the absolute deviation is no larger than $1/4$).

First, compare the polynomials of $\langle v_2, R_w(x)\rangle$ and $x_1^2$, we must have (1) $v_{2,1}w_1^2 \geq 3/4$ due to the $x_1^2$ term, and due to the $x_1x_2^2$ term, one has (2) $|v_{2,1}w_1w_2| \leq 1/4$. These two indicate (3) $|w_1| \geq 3|w_2|$. Then compare the polynomials of $\langle v_1, R_w(x)\rangle$ and $x_2^2$, we have (4) $|v_{1,1}w_1^2| \leq 1/4$ due to the $x_1^2$ term. Combining (3) and (4), one has (5) $|v_{1,1}w_2^2| \leq \frac{1}{9}|v_{1,1}w_1^2| \leq \frac{1}{36}$. Since the $x_2^2$ term is roughly matched, one must have (6) $|v_{1,2}w_1^2| \geq 1 - \frac{1}{4} - \frac{1}{36} = \frac{13}{18}$. However, note that (4) and (6) contradicts with the assumption that $|v_{1,1}| \geq |v_{1,2}|$. We conclude the proof. $\square$

We provide the proof of a technical Lemma used in proving Theorem 2.12

**Lemma C.1** (Technical tool). *Let $\Pi_n^d$ be the space of all polynomial of degree at most $d$ in $n$ variables. For any two polynomials $p_1(x), p_2(x) \in \Pi_3^2$, if*

$$\mathbb{E}_{x \sim \mathcal{B}_3(0,1)}[(p_1(x) - p_2(x))^2] \leq \frac{1}{1000},$$

*then the absolute deviation of each coefficient is at most $1/4$.*

*Proof.* Let $p(x) = (p_1(x) - p_2(x))^2$, taking an integral over $B_3(0, 1)$, we can only need to consider all quadratic terms, since all odd terms would be canceled due to symmetry. We divide into cases. (1) The coefficient of the constant term is greater than $1/4$, then $p(x) \geq 1/16$. (2) The coefficient of $x_1$ is greater than $1/4$, then $p(x) \geq \mathbb{E}_{x \sim \mathcal{B}_3(0,1)} \frac{1}{16}x_1^2 = \frac{1}{16} \cdot \frac{1}{5} = \frac{1}{80}$. (3) The coefficient of $x_1x_2$ is greater than $1/4$, then then $p(x) \geq \mathbb{E}_{x \sim \mathcal{B}_3(0,1)} \frac{1}{16}x_1^2x_2^2 = \frac{1}{16} \cdot \frac{1}{35} = \frac{1}{560}$. (4) The coefficient of $x_1^2$ is greater than $1/4$, then then $p(x) \geq \mathbb{E}_{x \sim \mathcal{B}_3(0,1)} \frac{1}{16}x_1^4 = \frac{1}{16} \cdot \frac{3}{35} = \frac{3}{560}$. Hence we conclude no coefficient has difference greater than $1/4$. $\square$

**Lower bound with ReLU activation**  There is nothing particularly special about the activation quadratic activation function: here, we provide a similar lower bound for features are represented via one-layer convolutional neural network with ReLU activation.

**Theorem C.2.** *Let $k, r, d \geq 2$. There exists a class of non-linear feature mappings and a sequence of environments, such that there is no (proper) continual learning algorithm that can guarantee to achieve less than $1/3$-error over all environments with probability at least $1/16$. The lower bound is constructed on a single family of two-layer neural network with ReLU activation.*

*Proof.* It suffices to take $k = 2, d = 2, r = 2$. The input distribution is uniform over $(1, 1), (-1, -1), (1, -1), (-1, 1)$ for both tasks. The hypothesis class $\mathcal{H}$ contains all two-layer convolutional neural network with a single kernel of size 1 and ReLU activation. That is, the representation function is parameterized by $w \in \mathbb{R}$ and $R_w(x) = (\max\{wx_1, 0\}, \max\{wx_2, 0\}) \in \mathbb{R}^2$ for input $x = (x_1, x_2) \in \mathbb{R}^2$.

The hard sequence of environment are drawn as follow: (1) The objective value in first environment is always $(0, 0, 1, -1)$; (2) The objective value of the second environment equals $(2, 0, 1, 1)$ with probability $1/2$ and $(0, 2, 1, 1)$ with probability $1/2$.

One can easily verify that the continual learning task is realizable: in the first case, one takes $w = 1$, $v_1 = (1, -1), v_2 = (1, 1)$ while in the second case, one takes $w = -1, v_1 = (-1, 1), v_2 = (1, 1)$.

We next prove any proper continual learning algorithm makes error at least $1/10$. We prove by contradiction and assume the continual learning algorithm takes value $w$ for the convolutional layer in the first task. It is easy to verify that $w \neq 0$, otherwise the first environment suffers loss at least $1/2$. When $w > 0$, then representation function equals $(w, w), (0, 0), (w, 0), (0, w)$ and we have $v_1 = (\frac{1}{w} \pm \frac{1}{2w}, -\frac{1}{w} \pm \frac{1}{2w}) \in (\mathbb{R}_+, \mathbb{R}_-)$. For the second environment, suppose the objective value equals $(0, 2, 1, 1)$ (note this happens with probability $1/2$). Then the algorithm must change the parameter to $w' < 0$, otherwise the loss on point $(-1, -1)$ is at least $4$. Then for the first task, the loss on point $(1, -1)$ is at least $1$. The case of $w < 0$ is similar and we conclude the proof here. $\square$

## D   Additional details of simulation

We provide further details for our simulations. In our simulations, we set input dimension $d = 100$, the number of features $r = 20$ and the continual learning setup uses $k = 500$ tasks. Each entry of the ground truth $U^\star \in \mathbb{R}^{d \times r}$ (and $V^\star \in \mathbb{R}^{k \times r}$) is drawn from the Gaussian $N(0, 1)$, and $W = U^\star(V^\star)^\top \in \mathbb{R}^{d \times k}$. The input data $x$ of each task is drawn from the multivariate gaussian $N(0, I_d)$, and the label is set to be $y = \langle w_i, x \rangle$. For each task, we drawn $N = 1000$ samples, and perform DPGrad/OGD/SGD for $T = 3000$ iterations, with learning rate $\eta = 0.01/0.0001/0.01$ respectively, and the initialization scale $\sigma = 0.01$. We omit the rounding step of DPGrad and simply takes $\widehat{w}_i = Uv_i$ (Line 12 in Algorithm 1), and the result of simulation empirically verifies that our Bit complexity assumption is indeed for convenience of analysis and one does not need it for practice. The OGD algorithm does not always converge in our simulations and we (1) decrease the learning rate and (2) perform early stopping: the projection is only w.r.t. the first 20 tasks. Our experiments are executed on an Apple M1 CPU.

## E   Additional Experiments

In addition to our synthetic data experiments, we also perform experiments on two common benchmark datasets: Permuted MNISTs and Rotated MNISTs. We do this both to verify the behavior of our proposed algorithm, as well as compare with two baseline approaches: Vanilla Stochastic Gradient Descent (SGD) and Orthogonal gradient descent (OGD) [11].

**Datasets**  We consider two datasets, Permuted MNIST and Rotated MNIST. In the Permuted MNIST dataset, a task is created by performing a random permutation to the input pixels; in the Rotated MNISTs a task is created by randomly rotating the input image. We generated 10 tasks for both benchmark datasets and the continual learning algorithm is sequentially exposed to these 10 tasks. The permutation/rotation is same within each task but different across tasks. Each task contains 60000 training samples and the test set contains 10000 images.

**Our methods**  Our DPGrad algorithm is tailored to the linear regression setting we consider, so it has to be modified to apply it to multi-class classification problems like Rotated/Permuted MNIST and/or to handle non-linear representations. We consider two natural generalizations of DPGrad.

To adapt it to a multi-class classification problem, we view each task as having 10 linear predictors—one for each class. Recall the key idea of DPGrad is to perform (fine-grained) column/row projection for the gradient of weight matrix and the column/row space is increased by (at most) 1 after each task. In the multi-class case, we force the increase of the row/column space to be (at most) 10 dimensions per task. In the tables/figures below we just call this DPGrad.

To adapt it to non-linear representations, we use a modified approach we call DPGrad+. In the linear setting, the column/row space increases by (at most) one dimension after each task and the newly added column/row is essentially the top eigenvector of the feature matrix $U$ as it is close to a rank-one matrix (see Lemma B.3) after projection. For non-linear feature, there is no reason to hope the weight matrix is rank-one, but instead, we perform singular value decomposition (SVD) to the matrix and take the top-$h$ eigenvectors and then add them to the column/row space. In other words, the only difference between DPGrad+ and DPGrad is that DPGrad+ augment the column/row space by the top-$h$ eigenvector instead of the top-1 eigenvector. We take $h = 15$ in both experiments.

**Hyperparameter choices**    We use a two-layer fully connected neural network, where the hidden layer contains 300 neurons and uses ReLU activation (for DPGrad+; for DPGrad the activation is linear). The parameters of the first layer are shared across tasks, while the weights of the second layer are different across tasks (i.e. the linear predictor). We perform 5 epochs of training for each task, the learning rate is fixed to be 0.1 and the batch size is 100.

**Experimental results**    The experimental results on Permuted MNIST can be found at Figure 3, Figure 5 and Table 4, the results on Rotated MNIST can be found at Figure 2, Figure 4 and Table 3. Figure 2-5 plot the test accuracy on the 10 tasks over time, Table 4 and Table 3 record the test accuracy of each tasks at the end of training (i.e., after the 10-th task). The average accuracy is reported in Table 1. The deviations of the values and confidence intervals are gotten from 5 runs, randomizing over the order of the tasks, as well as the randomness of the algorithm (i.e. seed).

Both DPGrad+ and DPGrad alleviate catastrophic forgetting and perform much better than vanilla SGD. Both outperform OGD, which is a strong baseline approach and outperforms classical approaches like elastic weight consolidation [19]. The performance of DPGrad+ and DPGrad is much more stable than OGD and the accuracy remains at a high level across tasks. By contrast, OGD has large variance across tasks—it obtains high accuracy in recent tasks but much lower accuracy in early tasks (especially in Rotated MNIST).

|          | Rotated MNIST        | Permuted MNIST       |
|----------|----------------------|----------------------|
| DPGrad+  | 76.6% ($\pm$2.1%)    | 89.5% ($\pm$0.2%)    |
| DPGrad   | 74.3% ($\pm$1.5%)    | 86.3% ($\pm$0.1%)    |
| OGD      | 73.0% ($\pm$2.4%)    | 88.7% ($\pm$0.6%)    |
| SGD      | 66.8% ($\pm$2.9%)    | 81.6% ($\pm$1.6%)    |

Table 1: Average Accuracy

Finally, we provide values for a common metric for quantifying forgetting: the *backward transfer value*, defined as

$$\frac{1}{k-1} \sum_{i=1}^{k-1} \mathsf{ACC}_{k,i} - \mathsf{ACC}_{i,i}$$

where $\mathsf{ACC}_{i,j}$ is the test accuracy of task $j$, after training with task $i$. A large negative backward transfer value means the algorithm suffers from catastrophic forgetting, a small or even positive backward transfer value indicates the algorithm avoids catastrophic forgetting. We report the backward transfer value in Table 2. In brief, OGD is more plastic than DPGrad, however at the expense of incurring a larger forgetting ratio (or negative backward transfer).

|          | Rotated MNIST      | Permuted MNIST     |
| -------- | ------------------ | ------------------ |
| DPGrad+  | -0.04 (±0.01)      | -0.01 (±0.01)      |
| DPGrad   | -0.01 (±0.01)      | 0 (±0.01)          |
| OGD      | -0.20 (±0.06)      | - 0.03 (±0.01)     |
| SGD      | -0.28 (±0.09)      | -0.10 (±0.05)      |

Table 2: Backward Transfer

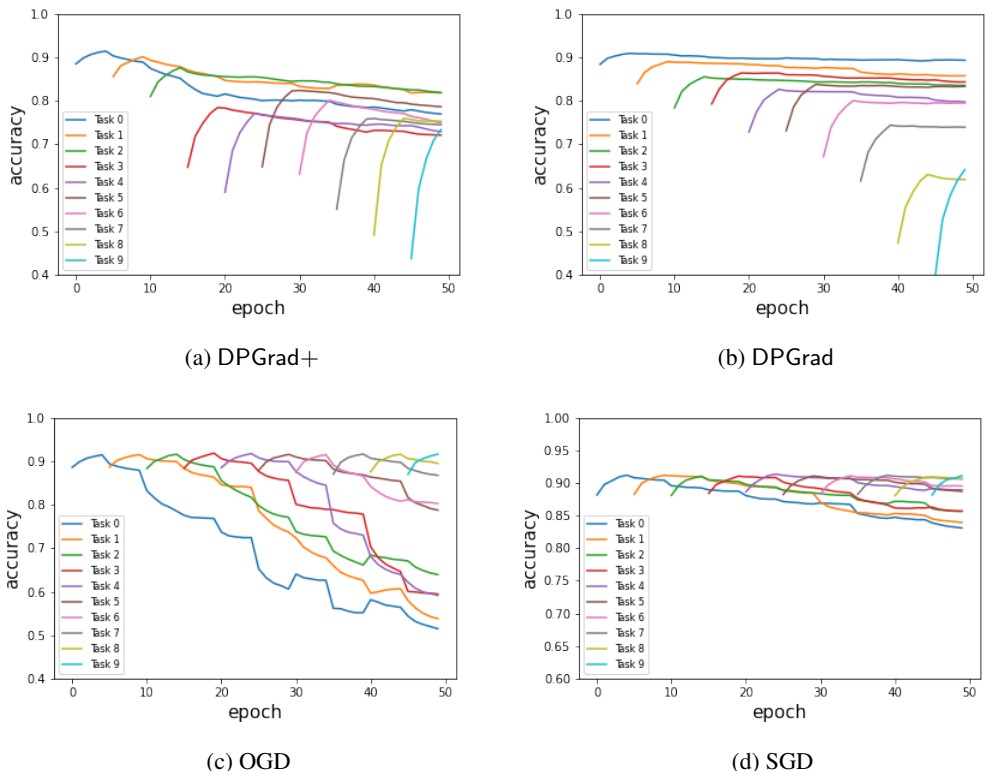

(a) DPGrad+  (b) DPGrad

(c) OGD  (d) SGD

Figure 2: Rotated MNIST

|          | Task 1           | Task 2            | Task 3            | Task 4            | Task 5            |
| -------- | ---------------- | ----------------- | ----------------- | ----------------- | ----------------- |
| DPGrad+  | 80.4% (±4.59%)   | 83.2% (±1.98%)    | 82.6% (±2.86%)    | 81.8% (±1.31%)    | 78.5% (±2.75%)    |
| DPGrad   | 82.6% (±1.19%)   | 79.5% (±1.46%)    | 77.3% (±6.02%)    | 76.6% (±2.97%)    | 76.3% (±2.85%)    |
| OGD      | 47.0% (±9.31%)   | 59.9% (±6.56%)    | 69.1% (±6.25%)    | 65.1% (±5.62%)    | 70.2% (±14.4%)    |
| SGD      | 49.7% (±9.37%)   | 52.8% (±12.8%)    | 56.5% (±14.9%)    | 55.3% (±15.0%)    | 63.5% (±8.45%)    |

|          | Task 6           | Task 7            | Task 8            | Task 9            | Task 10           |
| -------- | ---------------- | ----------------- | ----------------- | ----------------- | ----------------- |
| DPGrad+  | 74.8% (±9.24%)   | 71.7% (±4.68%)    | 73.2% (±3.27%)    | 70.6% (±4.79%)    | 69.5% (±5.22%)    |
| DPGrad   | 72.6% (±3.66%)   | 71.3% (±5.02%)    | 71.5% (±3.29%)    | 70.2% (±6.16%)    | 64.7% (±8.49%)    |
| OGD      | 77.0% (±4.80%)   | 83.6% (±4.73%)    | 78.9% (±5.64%)    | 87.6% (±2.33%)    | 91.7% (±0.29%)    |
| SGD      | 67.6% (±8.54%)   | 69.1% (±12.1%)    | 77.3% (±4.31%)    | 84.3% (±2.69%)    | 91.5% (±0.12%)    |

Table 3: Rotated MNIST

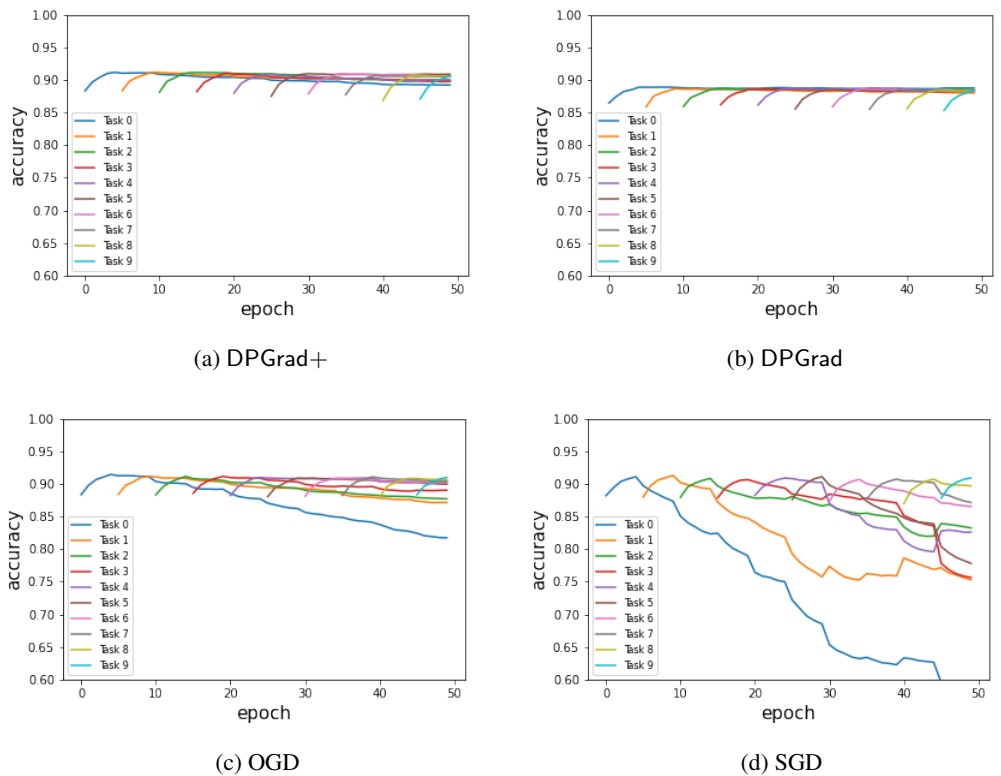

<table>
<tr><td></td><td>(a) DPGrad+</td><td></td><td>(b) DPGrad</td></tr>
<tr><td></td><td>(c) OGD</td><td></td><td>(d) SGD</td></tr>
</table>

Figure 3: Permuted MNIST

| | Task 1 | Task 2 | Task 3 | Task 4 | Task 5 |
|---|---|---|---|---|---|
| DPGrad+ | 87.1% ($\pm$2.14%) | 88.8% ($\pm$1.08%) | 87.9% ($\pm$1.12%) | 89.5% ($\pm$0.57%) | 89.2% ($\pm$1.32%) |
| DPGrad | 86.3% ($\pm$0.36%) | 86.1% ($\pm$0.64%) | 86.3% ($\pm$0.18%) | 86.3% ($\pm$0.36%) | 86.3% ($\pm$0.23%) |
| OGD | 82.9% ($\pm$4.37%) | 86.3% ($\pm$1.17%) | 88.2% ($\pm$1.05%) | 88.5% ($\pm$1.38%) | 89.2% ($\pm$0.59%) |
| SGD | 76.0% ($\pm$6.83%) | 67.9% ($\pm$9.03%) | 75.1% ($\pm$7.88%) | 78.6% ($\pm$3.85%) | 84.5% ($\pm$2.11%) |
| | Task 6 | Task 7 | Task 8 | Task 9 | Task 10 |
| DPGrad+ | 90.3% ($\pm$0.19%) | 90.5% ($\pm$0.16%) | 90.4% ($\pm$0.24%) | 90.4% ($\pm$0.11%) | 90.4% ($\pm$0.24%) |
| DPGrad | 86.2% ($\pm$0.32%) | 86.2% ($\pm$0.12%) | 86.3% ($\pm$0.05%) | 86.3% ($\pm$0.25%) | 86.3% ($\pm$0.13%) |
| OGD | 90.0% ($\pm$0.50%) | 90.3% ($\pm$0.50%) | 90.5% ($\pm$0.40%) | 90.6% ($\pm$0.13%) | 91.0% ($\pm$0.12%) |
| SGD | 82.9% ($\pm$7.05%) | 85.6% ($\pm$3.20%) | 86.6% ($\pm$2.33%) | 88.6% ($\pm$2.87%) | 90.7% ($\pm$0.24%) |

Table 4: Permuted MNIST

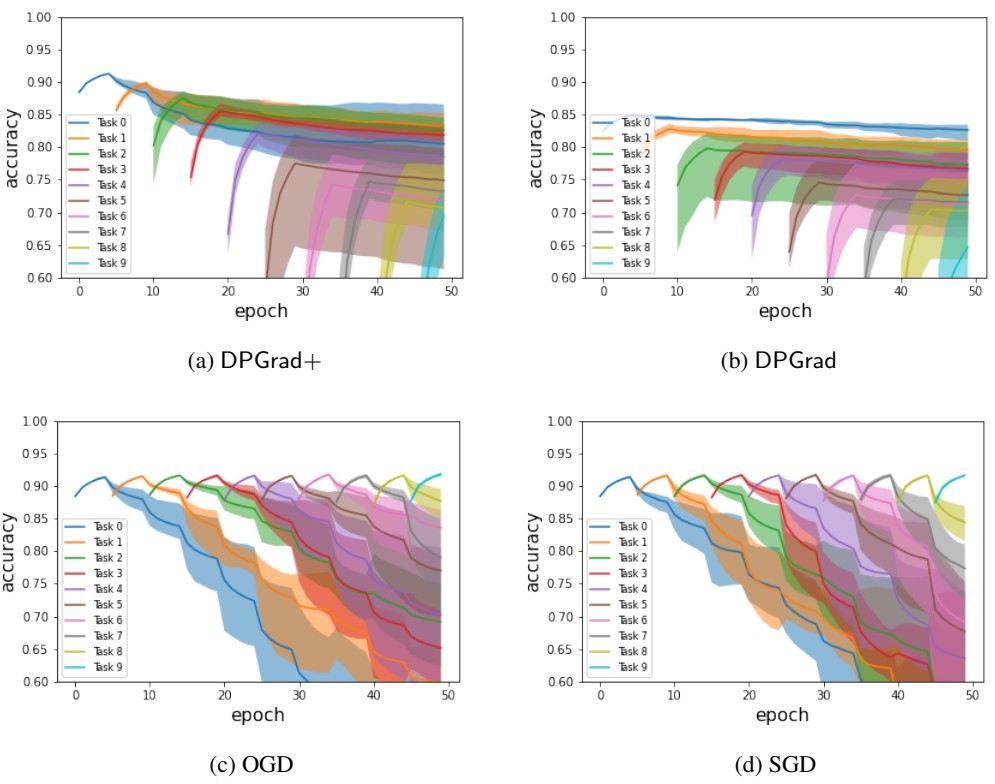

(a) DPGrad+            (b) DPGrad

(c) OGD            (d) SGD

Figure 4: Rotated MNIST (with error bar)

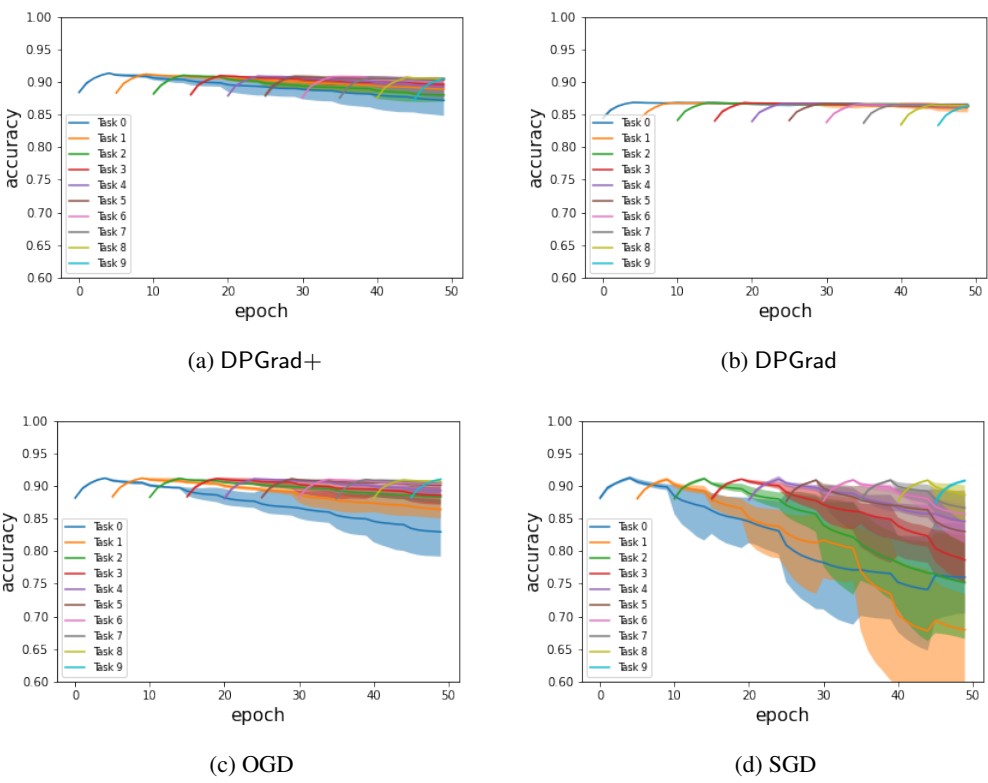

(a) DPGrad+            (b) DPGrad

(c) OGD            (d) SGD

Figure 5: Permuted MNIST (with error bar)