# OpenReview forum: "Continual learning: a feature extraction formalization, an efficient algorithm, and fundamental obstructions"
_NeurIPS.cc/2022/Conference — NeurIPS 2022 Accept_

### Official Review · Reviewer_KsmC · 2022-07-11

**Rating:** 6
**Confidence:** 3
**Soundness:** 3 good
**Presentation:** 3 good
**Contribution:** 2 fair

**Summary:**

The authors introduce DPGrad, an algorithm for task-incremental continual learning focused on the case where the features extractors are assumed to be linear. They also provide a lower bound on the error for a specific setting with non-linear features extractors.
DPGrad is compared with SGD and OGD on a simulated environment.

**Questions:**

* It would be great if the authors could provide more empirical evidence of the performance of their algorithm (maybe on standard benchmarks)
* See the other points in the paragraph above.

**Limitations:**

--

**Strengths And Weaknesses:**

Strengths:
* The paper rigorously defines the setting and assumptions before introducing the new solution
* The work in this paper tries to provide additional understanding of the theoretical aspects involved in the continual learning process. A direction which is heavily unexplored in the CL community due to the difficulty of the topic.

Weaknesses:
* The authors generically refer to continual learning in several parts of the paper but, as they correctly point out, there are several different settings in continual learning and their findings do not apply to all of them. It is particular important to be clear about it when providing negative results such the lower bound in Sec. 5.
* The definition of forgetting on L102 works because of the assumption that the learner will get a good accuracy on the current task (L103) but it would be wrong otherwise. Generally speaking, it would be more intuitive to see it expressed as a difference with the performance measured in the past on an observed task.
* It is quite hard to understand the potential practical impact of the findings in the paper and the results in Sec. 6 do not help. It is clear that the main contribution of this work is theoretical and the aim is not to outperform some continual learning baselines, but it would be beneficial for the community to understand how big is the gap between theory and practice.
* Sec 5 provides a lower bound on the error but it is not clear how much of a practical impact that is going to have. It would be interesting if the authors could discuss which realistic assumptions on the environments could provide a more favorable situation.

---

> ### Author Response · Authors · 2022-08-02
> **Reply to Reviewer KsmC**
>
> Thank you for the valuable and encouraging comments! We address your major concerns now:
>
>
> >- “The authors generically refer to continual learning in several parts of the paper but, as they correctly point out, there are several different settings in continual learning and their findings do not apply to all of them. It is particular important to be clear about it when providing negative results such the lower bound in Sec. 5.”
>
> Thanks for bringing this up—we tried to be very forceful about this point as well. We mentioned it one more time in the theorem statement of the lower bound as well to be even more explicit (e.g. see the new Theorem 2.12 in the updated draft)
>
> >- The definition of forgetting on L102 works because of the assumption that the learner will get a good accuracy on the current task (L103) but it would be wrong otherwise. Generally speaking, it would be more intuitive to see it expressed as a difference with the performance measured in the past on an observed task.
>
> Thanks for pointing this out, we agree with it. Our results were mostly focused on the “realizable setting” — that is, where there is a ground-truth featurizer with corresponding linear predictors that achieves 0 loss. In this case, these two notions are equivalent—but we agree that when there’s model mismatch the above distinction would be valuable.
>
> >- Sec 5 provides a lower bound on the error but it is not clear how much of a practical impact that is going to have. It would be interesting if the authors could discuss which realistic assumptions on the environments could provide a more favorable situation.
>
> Thanks for the suggestion! This is indeed one of the most interesting directions for further research: finding assumptions that make the problem tractable (i.e. circumvent known lower bounds), and designing algorithms under such assumptions. We can only speculate, but one reasonable assumption that would avoid our lower bound is some kind of “gradualness” in the change of the classifiers from one environment to the next (as opposed to an arbitrary drift).
>
> >- It is quite hard to understand the potential practical impact of the findings in the paper and the results in Sec. 6 do not help. It is clear that the main contribution of this work is theoretical and the aim is not to outperform some continual learning baselines, but it would be beneficial for the community to understand how big is the gap between theory and practice.
>
> As part of the rebuttal, we provide empirical results for two algorithms inspired by our theory in Appendix E, added to our updated draft. We provide results on two common benchmarks, Permuted MNIST and Rotated MNIST for two variants of our algorithm — one is a modification of DPGrad for *multi-class classification*, the other is a modification that allows *non-linear featurizers*. To adapt it to a multi-class classification problem, we view each task as having 10 linear predictors---one for each class. Recall the key idea of DPGrad is to perform (fine-grained) column/row projection for the gradient of the weight matrix and the column/row space is increased by (at most) 1 after each task. In the multi-class case, we force the increase of the row/column space to be (at most) 10 dimensions per task. To adapt it to *non-linear representations*, we note that in the linear case, the column/row space increases by (at most) one dimension after each task and the newly added column/row is essentially the top eigenvector of the feature matrix as it’s close to rank 1. For non-linear features, there is no reason to hope the weight matrix is rank-one, but instead, we perform singular value decomposition (SVD) to the matrix and take the top-h eigenvectors and then add them to the column/row space for some h.
>
> Detailed numbers and figures can be found in Appendix E in the updated draft. In brief, both algorithms alleviate catastrophic forgetting and perform much better than vanilla SGD. Both outperform OGD, which is a strong baseline approach. Furthermore, the performance of both is much more stable than OGD and the accuracy remains at a high level across tasks. By contrast, OGD has large variance across tasks—it obtains high accuracy in recent tasks but much lower accuracy in early tasks (especially in Rotated MNIST).

---

> > ### Comment · Reviewer_KsmC · 2022-08-05
> > **Thank you**
> >
> > After reading your answer I still have some concerns on the applicability of these results, but I appreciate the effort you made to provide more insights.

---

### Official Review · Reviewer_VAS6 · 2022-07-12

**Rating:** 6
**Confidence:** 3
**Soundness:** 4 excellent
**Presentation:** 3 good
**Contribution:** 3 good

**Summary:**

In the recent past, there have been several advancements in the field of Continual Learning (CL), where a big chunk of it is devoted to empirical research, such as the usage of replay/regularization-based methods. However, there has been much lesser attention to the theoretical aspects. This work considers continual learning in the regime of regression tasks, where the task descriptors are available. With the standard definition of catastrophic forgetting, the paper demonstrates an algorithm that provably avoids catastrophic forgetting, when the underlying data generating process is such that when the features that lead to optimum solutions to regression can be written as a linear mapping of inputs. Under this setting, they propose an algorithm called DPGrad, which is based on a projected gradient descent algorithm. They further show using a special case that in a non-linear regime, it is not possible to prevent catastrophic forgetting. Finally, in a synthetic toy example, they show the efficacy of DPGrad against OGD and plain SGD.


**Questions:**

As mentioned in the main review.

**Limitations:**

As mentioned in the main review.

**Strengths And Weaknesses:**

Strengths:

- Theoretical contributions are strong, even for the simplest case, that is linear model regime and squared loss, and proofs seem rigorous (although I’ve not completely verified them).


Weakness:

While it is primarily a theoretical work, much of the weaknesses of this work are related to limited applicability to the empirical setting. In the following, I will list down my questions:

- In Algorithm 1 Line 13, V is defined as the span of v_i. However in other places as well V is used, but maybe it is referring to row space (see line 205). This is getting a bit confusing to me.
- While a downside of having a memory-based replay method is mentioned in line 48, they are currently SOTA. How does DPGrad change when CL is accompanied by experience replay?
- The setting uses task descriptors and have different linear classifier for different tasks ($v_i$). However, again from a practical point of view it is a big assumption that much previous CL work (albeit empirical) tends to avoid. Can it be easily proven/disproven that even in a linear feature case if one uses a single head (just one v_i for all the tasks) catastrophic forgetting will happen? Intuitively it seems it cannot prevent forgetting.
- While the focus of this work is regression, what about the classification problems? Some of the previous work such as Gradient Episodic Memory (GEM) has the same formulation of no forgetting (assuming memory is a decent representation of task distribution) but uses local linear assumptions to provide a working algorithm. Can authors add some discussion on extending this to classification problems?
- Can the authors mention the following paper in the related work- “Reconciling meta-learning and continual learning with online mixtures of tasks” especially their section 6.1 have additional simulation-based datasets that can be used in this setting to improve the experiments section?

---

> ### Author Response · Authors · 2022-08-02
> **Reply to Reviewer VAS6**
>
> Thanks for the valuable comments. We are glad you thought the “theoretical contributions are strong”! We address your concerns below.
>
> >- While a downside of having a memory-based replay method is mentioned in line 48, they are currently SOTA. How does DPGrad change when CL is accompanied by experience replay?
>
> Part of the program we propose in this paper is that in order to understand what is **fundamentally possible and impossible** in different *settings of practical interest* for continual learning, one needs to formalize the intuitive considerations of the setting into a learning problem.
>
> Through this lens, it’s not surprising that memory replay helps: it is a fundamentally more permissive setting, in which the algorithm is allowed to store examples from prior tasks. This formalization wouldn’t be appropriate, for example, in situations where memory or privacy is a concern.) It would certainly be a very interesting research direction to explore fundamental tradeoffs in this setting as well (e.g. *how much* memory is needed to achieve a certain accuracy? Can a *differentially-private* algorithm achieve a certain accuracy?)
>
> >- The setting uses task descriptors and have different linear classifier for different tasks vi,  However, again from a practical point of view it is a big assumption that much previous CL work (albeit empirical) tends to avoid. Can it be easily proven/disproven that even in a linear feature case if one uses a single head (just one v_i for all the tasks) catastrophic forgetting will happen? Intuitively it seems it cannot prevent forgetting.
>
> Yes, in our set up, it is very easy to prove that when only one head is used, catastrophic forgetting can not be prevented.
>
> >- While the focus of this work is regression, what about the classification problems? Some of the previous work such as Gradient Episodic Memory (GEM) has the same formulation of no forgetting (assuming memory is a decent representation of task distribution) but uses local linear assumptions to provide a working algorithm. Can authors add some discussion on extending this to classification problems?
>
> There is nothing in our approach that is inherently tied to regression. The formalism we consider (Definition 2.2) is in fact not specific to regression: the labels y can just as easily be discrete (and the loss can of course be changed as well). We suspect our positive results can be extended to classification as well.
>
> >- Can the authors mention the following paper in the related work- “Reconciling meta-learning and continual learning with online mixtures of tasks” especially their section 6.1 have additional simulation-based datasets that can be used in this setting to improve the experiments section?
>
> Thanks for bringing this up! We have added the reference to the related works. We ran some additional experiments on Permuted MNIST and Rotated MNIST (See the added Appendix E to the updated draft), but due to time constraints, we didn’t have the time to also run experiments on these datasets. We are happy to do it for the final version of the paper.

---

> > ### Comment · Reviewer_VAS6 · 2022-08-06
> > **Response to the authors.**
> >
> > Thanks for providing a way to extend the work to the classification setting, and adding experiments for the common datasets. I have a few comments:
> >
> > 1. Regarding the choice of *h* for the number of top eigenvectors used for the algorithm, I am suspecting this is decided based on the eigenspectrum. If so, any intuitive explanation of why the same h worked well for both the datasets?
> >
> > 2. For the Rotated MNIST dataset, from Table 2 it seems that OGD outperforms every method from task 6 onwards (quite significantly for the later tasks) pointing out that DPGrad is not plastic about learning new knowledge at later stages. Is it possible for the authors to include metrics such as backward transfer to quantify forgetting, which is very common metric in Continual Learning literature?
> >
> > 3. For the Rotated MNIST dataset, how does the task ordering affect the performance? I am suspecting that only one task order is used in the reported results, but I am curious to see what happens if the results are averaged across task orders and also, different random initializations. Checklist 3c states that results don't report the error bars because of a large margin, but I think in classification experiments authors should add error bars.

---

> > > ### Author Response · Authors · 2022-08-08
> > > **Reply to Reviewer VAS6**
> > >
> > > Thank you for the reply and comments! We address your questions below.
> > >
> > > >- Regarding the choice of h for the number of top eigenvectors used for the algorithm, I am suspecting this is decided based on the eigenspectrum. If so, any intuitive explanation of why the same h worked well for both the datasets?
> > >
> > > For the experiments we provided results for, we choose the same h across all tasks on both datasets and its value is tuned on the Rotated MNIST. One intuitive explanation why the same h might work for both datasets (Permuted MNIST and Rotated MNIST) is that both dataset/tasks are variants of MNISTs (up to the application of the permutation/rotation), so the learned feature space could be quite similar.
> > >
> > >
> > > >-  For the Rotated MNIST dataset, from Table 2 it seems that OGD outperforms every method from task 6 onwards (quite significantly for the later tasks) pointing out that DPGrad is not plastic about learning new knowledge at later stages. Is it possible for the authors to include metrics such as backward transfer to quantify forgetting, which is very common metric in Continual Learning literature?
> > >
> > > We have added the backward transfer values in Appendix E, Table 3. Indeed, we agree that OGD seems to be more plastic than DPGrad, however at the expense of incurring a larger forgetting ratio (or negative backward transfer).
> > >
> > >
> > > >- For the Rotated MNIST dataset, how does the task ordering affect the performance? I am suspecting that only one task order is used in the reported results, but I am curious to see what happens if the results are averaged across task orders and also, different random initializations. Checklist 3c states that results don't report the error bars because of a large margin, but I think in classification experiments authors should add error b.
> > >
> > > Thank you for the question! We have now added the maximum deviations and confidence bars for both the accuracies and the backward transfer, averaged over 5 runs randomized for ordering of the tasks and random seed. The updated tables (Tables 2,3,4) and figures (Figures 4,5) can be found in Appendix E. We are happy to average over more runs, but due to the time constraints of the rebuttal period, this is as much as we could run. We find all the quantities are relatively stable — especially the average accuracy.

---

> > > > ### Comment · Reviewer_VAS6 · 2022-08-08
> > > > **Re: On experiments.**
> > > >
> > > > Thanks for the clarifications!  Indeed OGD seems to have a massive forgetting. Looking at the results with a variety of task orders, I have a little concern about this trade-off of plasticity v/s learnability of new tasks, which is in general a big question in the CL community. This problem is in general alleviated to some extent in replay-based methods, however, I cannot use them as the baseline numbers while assessing the validity of the provided numbers since the setting is altogether different. Thanks for the experiments and for providing a theory for the linearly separable case. Lastly, I've decided to keep my rating (6).

---

### Official Review · Reviewer_cwgv · 2022-07-14

**Rating:** 5
**Confidence:** 4
**Soundness:** 2 fair
**Presentation:** 3 good
**Contribution:** 2 fair

**Summary:**

The authors consider defining a theoretical framework allowing to study the continual learning problem. They look at this problem from the point of view of feature extraction, where features are continually learned on a sequence of environments, followed by task specific linear classifiers.
They then provide an analysis of two cases: the case where the features are linear, and the case where they are not. In the first case, they provide an algorithm that is (under certain well defined assumptions) guaranteed to converge and that gives a learner that does equally well on the past and on the present tasks. In the second case, they construct a counterexample on which no continual learning can guarantee a low error with high enough probability.
The algorithm provided for the first case is validated by a simulation on synthetic data, and compared to vanilla stochastic gradient descent and to orthogonal gradient descent from which the proposed algorithm is heavily inspired.

**Questions:**

I think the paper can be stronger by clarifying certain points.

- The authors recognize the similarity between their proposed algorithm and the gradient projection family of methods that have been previously proposed. Can they state more clearly the differences and why their proposed method is a novel one?

- While the feature extraction take on continual learning is interesting, it would be more interesting in practice to study the case where the feature extractor and the linear classifier are trained simultaneously, taking it closer to e.g. neural networks training. Can the current analysis be extended to cover this case?

- The study of the non-linear case is based on a quadratic activation function and a polynomial target. Most of the neural networks used nowadays use ReLU activations or its variants, and encode piecewise linear functions. Can this fact change the analysis of the non-linear case, and do the results change in that case, especially when coupled with a joint training of the features and the linear classifiers?


**Limitations:**

The paper is mostly theoretical, and a study of societal impact seems out of scope here.
The work can however be improved by bringing the non-linear case study closer to settings used in practice. This can lead to a much higher impact and improved understanding of continual learning.

**Strengths And Weaknesses:**

Strengths:
* The paper is looking at filling a gap in the literature between theory and practice. It is looking at a problem that can be very useful for the community to improve our understanding of Continual Learning in general.
* The paper is well written, with a clear structure and easy to follow.
* Looking at the problem from a feature extractor point of view is quite intuitive for the supervised setting that is the focus here.
* The analysis of the linear case is very detailed and clear, and the results in this case are sound and interesting.

Weaknesses:
* While the paper is well structured, it seems quite unbalanced between the linear case and the non linear one. It is focused mostly around the former, while the latter is more interesting in practice. The motivation of the authors is that the linear case provides intuition for the non-linear case. This is not generally true. From a neural network perspective for example, this can hold for the overparameterized setting, in which the input has a significantly lower dimension than the features. The analysis conducted in this paper consider the opposite setting with low dimensional features, and the insights derived from the linear case are likely to break for the non-linear case.
* The conclusion for the non-linear case seems to stretch the argument more than it allows. The considered setting is so particular and little used in practice (especially for the supervised learning setting). In particular, the choice of the activation function and the changing objective between the environments are not used in practice. Moreover, the choice of learning the features and the linear classifier separately can highly impact the results. Showing that there is no proper continual learner with sufficiently low error in this setting seems insufficient to conclude that it is the case in general for the non-linear case.
* The experimental results are interesting as they mimic exactly the theorem assumptions. They are however very limited. It would be interesting to conduct some experiments with real data, even at a small scale (where a linear model can already give reasonable performance), to see if the observations still holds when the assumptions are partly relaxed.

---

> ### Author Response · Authors · 2022-08-02
> **Reply to Reviewer cwgv**
>
> Thanks for the valuable comments. We are glad you found our feature-extractor formalism “intuitive”, the analysis of the linear case “interesting” and the paper overall “well written” and “easy to follow” ! We think most of the concerns come from a slight misunderstanding of our formalization, and hopefully our clarification below will assuage these worries.
>
> >- You asked: “it would be more interesting in practice to study the case where the feature extractor and the linear classifier are trained simultaneously, taking it closer to e.g. neural networks training. Can the current analysis be extended to cover this case?”
>
> **The extractor and classifier can indeed be simultaneously trained.** Typical algorithms — as well as our algorithm for the linear case— will indeed train them jointly using some form of (possibly regularized, or projected) gradient descent. The only restriction in our setting is that for *past* tasks, the weights of the linear classifier cannot be updated as the feature extractor is updated. The motivation behind this is that to do so, one would need to retrain on the past examples — which often we don’t have access to due to memory or privacy concerns.
>
> >- You asked: “The study of the non-linear case is based on a quadratic activation function and a polynomial target. Most of the neural networks used nowadays use ReLU activations or its variants, and encode piecewise linear functions. Can this fact change the analysis of the non-linear case, and do the results change in that case, especially when coupled with a joint training of the features and the linear classifiers”
>
> Joint training is already allowed, as we explained above. ReLU activation is not an essential ingredient to the lower bound — we can construct a similar example for ReLU activations. We have added this construction as Section C, Theorem C.2 in the updated draft.
>
> >- You asked: “The authors recognize the similarity between their proposed algorithm and the gradient projection family of methods that have been previously proposed. Can they state more clearly the differences and why their proposed method is a novel one”
>
> The gradient projection family of methods (taking OGD for example) view the entire feature matrix as a vector (of dimension dr) and perform projection on it. Our algorithm, instead, projects twice — with respect to the column and row space, separately. The fine-grained projection is the novel part and it is crucial.
>
>
> >- You asked: “The experimental results are interesting as they mimic exactly the theorem assumptions. They are however very limited. It would be interesting to conduct some experiments with real data, even at a small scale (where a linear model can already give reasonable performance), to see if the observations still holds when the assumptions are partly relaxed.”
>
> As part of the rebuttal, we provide empirical results for two algorithms inspired by our theory in Appendix E, added to our updated draft. We provide results on two common benchmarks, Permuted MNIST and Rotated MNIST for two variants of our algorithm — one is a modification of DPGrad for *multi-class classification*, the other is a modification that allows *non-linear featurizers*. To adapt it to a multi-class classification problem, we view each task as having 10 linear predictors---one for each class. Recall the key idea of DPGrad is to perform (fine-grained) column/row projection for the gradient of the weight matrix and the column/row space is increased by (at most) 1 after each task. In the multi-class case, we force the increase of the row/column space to be (at most) 10 dimensions per task. To adapt it to *non-linear representations*, we note that in the linear case, the column/row space increases by (at most) one dimension after each task and the newly added column/row is essentially the top eigenvector of the feature matrix as it’s close to rank 1. For non-linear features, there is no reason to hope the weight matrix is rank-one, but instead, we perform singular value decomposition (SVD) to the matrix and take the top-h eigenvectors and then add them to the column/row space for some h.
>
> Detailed numbers and figures can be found in Appendix E in the updated draft. In brief, both algorithms alleviate catastrophic forgetting and perform much better than vanilla SGD. Both outperform OGD, which is a strong baseline approach. Furthermore, the performance of both is much more stable than OGD and the accuracy remains at a high level across tasks. By contrast, OGD has large variance across tasks—it obtains high accuracy in recent tasks but much lower accuracy in early tasks (especially in Rotated MNIST).

---

> ### Author Response · Authors · 2022-08-08
> **Gentle reminder for Reviewer cwgv**
>
> Dear Reviewer cwgv,
>
> thank you for putting in the time and effort to review our paper.
>
> We hope our clarification resolved your concerns: in particular, we clarified that the extractor and classifier are indeed trained together in our framework; we also added a version of the lower bound for ReLU activation to demonstrate the lower bound isn't sensitive to the activation function (Section C, Theorem C.2 in); finally, we added substantially more experiments in Appendix E on more realistic datasets (Rotated and Permuted MNIST).
>
> We'd love to hear back. If we've sufficiently addressed your concern, could you please reevaluate our score? Thank you very much!

---

> > ### Comment · Reviewer_cwgv · 2022-08-08
> > **Thanks for the answers  - Correct thread**
> >
> > Thanks for your quick reaction, and apologies for posting in the wrong thread.
> >
> > To be more precise, I was thinking about settings where the drift is less abrupt between environments. It seems to me for example that the shift in objective studied in the non-linear case is not common in practice, at least not in classification settings that are the focus here, and I am therefore unclear how much the insights gained with this study would hold in more common environments.
> >
> > This being said, I have seen in the answer to another reviewer that this goes beyond the scope of this work and is considered as a potential future research direction. I therefore increased my score accordingly as mentioned in my previous comment.

---

### Official Review · Reviewer_s7RD · 2022-07-18

**Rating:** 6
**Confidence:** 3
**Soundness:** 3 good
**Presentation:** 2 fair
**Contribution:** 2 fair

**Summary:**

The papers considers the incremental problem setting of Continual Learning (CL) and investigates the performance of methods with a multi-head architecture. This setup is seen through the lens of feature extraction, in which there is a shared network which extracts features from the input, and a task-specific linear classifier. The paper reviews two cases. In the first, the feature extractor is a linear mapping of the input. For this case, the authors present a gradient-based algorithm which is guaranteed to be able to learn to perform new tasks, while avoiding catastrophic forgetting. In the second case, the feature extractor is a nonlinear mapping of the input. The authors show that there is a CL sequence in which a CL algorithm cannot reliably achieve a high performance.

**Questions:**

q1) What reasons would you give that this paper provides an important step forward?
q2) How would you compare your results in section 5 to the conclusions drawn in [1]?
q3) On line 338, do you mean “r=3” instead of “n=3”?


**Limitations:**

I did not find a discussion on the limitations of the current findings.


**Strengths And Weaknesses:**

I think that this paper constitutes a step in a very interesting direction, namely, exploring the limitations of different CL approaches. I also think that the contributions made in the paper are novel.
However, I am unsure about the paper’s significance. Most of the presentation is dedicated to developing an algorithm which works well when the target feature map is linear. However, there is no discussion on how the insights gained from this are actually relevant to the non-linear setting. Therefore, I am left with the impression that this cannot be used to further the theoretical research in Continual Learning.
I found the result for non-linear features to be interesting, however, there is a missing discussion on why this is an impactful result. What is more, I think that [1] already contains a similar result, while reached in a different way. Concretely, that if a CL algorithm does not store any data points of past tasks, it’s not going to achieve optimal performance (please correct me if I’m wrong).
In terms of clarity, I do not think that the current state of CL research is correctly depicted in the paper. For instance in paragraph 28-34, the authors discuss only a subset of the CL ideas  (a domain-incremental CL setting with regularisation-based approaches) while claiming to summarise all of CL. Moreover, I think that the proof sketch in Section 5 needs another iteration in order to improve readability.

[1] “Optimal Continual Learning has Perfect Memory and is NP-HARD” , Jeremias Knoblauch, Hisham Husain, Tom Diethe

---

> ### Author Response · Authors · 2022-08-02
> **Reply to Reviewer s7RD (Continued)**
>
> >- How would you compare your results in section 5 to the conclusions drawn in [1].
>
> Thanks for bringing this up! The settings in the two papers are quite different, and the results are (even qualitatively) very different. In brief, the results in [1] have a substantially more “worst case” flavor than ours. For instance, given the generality of their formalization, the NP-hardness results are not surprising: optimization problems with arbitrary distributions/predictor classes are typically NP-hard in the worst case; by contrast, our results, as we mentioned above, show that there are fundamental obstructions due to the online nature of the setting — *not* just due to computational nor statistical constraints. Their memory lower bounds are based on a kind of “counting” argument and when translated to our setting, would just say that $\Omega(dr)$ on the memory size is needed, which is not interesting since we already have dr parameters in the feature function. These differences stem from the fact that we consider a more fine-grained family of algorithms based on featurizers with a separate linear predictor (Definition 2.2). We’ve added some discussion to this effect in the updated draft, Appendix A.
>
> >- “On line 338, do you mean “r=3” instead of “n=3”?”
>
> Good catch! Indeed, we mean k=2, d = 3, r=2.
>
>
> >- For instance in paragraph 28-34, the authors discuss only a subset of the CL ideas (a domain-incremental CL setting with regularization-based approaches) while claiming to summarise all of CL.
>
> Thanks for pointing this out. Due to space constraints, we included a more detailed discussion of CL broadly in Appendix A, where we discuss other CL settings and approaches (e.g. regularization based-approaches, memory replay, dynamic architectures). We have moved some of this discussion back to the main body in the updated draft.

---

> ### Author Response · Authors · 2022-08-02
> **Reply to Reviewer s7RD**
>
> Thanks for the valuable comments. We are glad you found our paper constitutes “step in a very interesting direction” and that the contributions are “novel”. Your concerns seem to mostly revolve around the significance of the paper and how it provides a “step forward” which we hopefully will alleviate with this reply.
>
> **Relevance of insights and why the paper is an important step forward**: Thank you for this question. We think both the algorithm for the linear setting and the lower bound in the nonlinear setting offer useful insights, as well as directions for further work.
>
> The algorithm in the *linear setting* provides an insight into projection-based methods like Orthogonal Gradient Descent (OGD) [Farajtabar et al, 2019] — and simultaneously show that the way the projection should be chosen can be subtle (i.e. we need to project both on the left and right, i.e. the row and column space). It would certainly be very interesting to prove analogous results beyond linear featurizers—though we point out that understanding gradient-descent like algorithms even for the basic, supervised learning setting for non-linear networks is essentially wide open except in some restricted settings like NTK (very wide networks).
>
> The significance of the *lower bounds* is as part of a program to delineate what is **fundamentally possible and impossible** in different formalizations for continual learning. In the feature-based formalization we consider (which makes sense when memory is an issue, both for growing the model, and privacy for retaining a memory buffer of prior examples) — our lower bound shows that continual learning for general distributions and non-linear classifiers is impossible. Moreover, this is not due to computational constraints (i.e. NP-hardness) or statistical constraints (i.e. large sample complexity) — but a fundamental obstruction due to the online nature of the setting. Precisely, in some task, it’s possible that multiple pairs of (featurizer, classifier) work, but some featurizers will not be good for the subsequent tasks.
>
> As part of the rebuttal, we provide empirical results for two algorithms inspired by our theory in Appendix E, added to our updated draft. We provide results on two common benchmarks, Permuted MNIST and Rotated MNIST for two variants of our algorithm — one is a modification of DPGrad for *multi-class classification*, the other is a modification that allows *non-linear featurizers*. To adapt it to a multi-class classification problem, we view each task as having 10 linear predictors---one for each class. Recall the key idea of DPGrad is to perform (fine-grained) column/row projection for the gradient of the weight matrix and the column/row space is increased by (at most) 1 after each task. In the multi-class case, we force the increase of the row/column space to be (at most) 10 dimensions per task. To adapt it to *non-linear representations*, we note that in the linear case, the column/row space increases by (at most) one dimension after each task and the newly added column/row is essentially the top eigenvector of the feature matrix as it’s close to rank 1. For non-linear features, there is no reason to hope the weight matrix is rank-one, but instead, we perform singular value decomposition (SVD) to the matrix and take the top-h eigenvectors and then add them to the column/row space for some h.
>
> Detailed numbers and figures can be found in Appendix E in the updated draft. In brief, both algorithms alleviate catastrophic forgetting and perform much better than vanilla SGD. Both outperform OGD, which is a strong baseline approach. Furthermore, the performance of both is much more stable than OGD and the accuracy remains at a high level across tasks. By contrast, OGD has large variance across tasks—it obtains high accuracy in recent tasks but much lower accuracy in early tasks (especially in Rotated MNIST).

---

> ### Author Response · Authors · 2022-08-08
> **Gentle reminder for reviewer s7RD**
>
> Dear Reviewer s7RD,
>
> thank you for putting in the time and effort to review our paper.
>
> We hope our clarification resolved your concerns: in particular, we clarified how our results fit in a broader, ambitious program to understand the possibilities and limitations of different formalizations of continual learning; we also added a substantial number of new experimental results in Appendix E on more realistic datasets (Permuted and Rotated MNIST); finally, we clarified how our results qualitatively differ from [1] --- both in terms of the framework and the implications of the results.
>
> We'd love to hear back. If we've sufficiently addressed your concern, could you please reevaluate our score? Thank you very much!

---

> > ### Comment · Reviewer_cwgv · 2022-08-08
> > **Thanks for the answers**
> >
> > I thank the authors for their clarifications.
> >
> > While certain aspects of the contributions are made clearer, it is still unclear to me how they relate to more realistic settings.
> > I increased my score accordingly.

---

> > > ### Author Response · Authors · 2022-08-08
> > > **Re: Thanks for the answers**
> > >
> > > Thank you as well!
> > >
> > > With respect to "more realistic settings": are there some specific datasets or metrics you would like to see? The time until the end of the discussion period is very limited, but we can do our best to try to provide some results to the extent time permits. Similarly, if there are outstanding questions about the theory, we're happy to answer them.
> > >
> > > (P.S. This reply somehow ended up in the thread of reviewer s7RD. We're happy to continue here, or we can switch to your (cwgv) thread.)

---

### Meta-Review · Area_Chair_t5i7 · 2022-08-27

**Recommendation:** Accept
**Confidence:** Certain

**Metareview:**

This paper provides a theoretical analysis of continual learning when the learner is modeled as a featurizer followed by a linear head. The analysis provides theoretical guarantees on learnability when the featurizer is linear, and is learned using doubly projected gradient descent. The guarantee ensures good accuracy on all environments and resilience to catastrophic forgetting. For a nonlinear featureizer, it is shown that continual learning is not possible in general: there exist scenarios that even when good features exist, either catastrophic forgetting or poor performance has to occur.

Reviewers raised questions about the implication of the theory for practical settings (s7RD, KsmC about how useful the results are in practice, cwgv on whether current analysis based on quadratic activation functions can carry over to ReLU, and VAS6 about whether the analysis works for classification as well [instead of regression]). The authors responded to these. They highlighted  how the algorithm in the linear setting provides an insight into Orthogonal Gradient Descent (OGD), which is known algorithm for continual learning. Authors also explained the significance of the lower bounds for understanding what is fundamentally possible and impossible in continual learning. Moreover, the authors clarified that quadratic activation is not essential for the lower bound, and extended their proof to ReLU in the revised version of the submission. The authors also clarified that classification could be treated as a special case of regression, with target values being discrete, and hence the result is not limited to regression only (although I, the AC, add that for classification, L2 loss is less common, and by classification, we typically refer to loss functions like cross-entropy, but the results of the paper are interesting regardless).

Final scores are all on the accept side, indicating reviewers have found the contributions strong enough for the submission to be published. In concordance, I think this paper provides very interesting insights about some fundamental aspects of learnability in continual settings.


**Award:**

No

---

### Decision · Program_Chairs · 2022-09-14

Accept